# Long-term treatment with senolytic drugs Dasatinib and Quercetin ameliorates age-dependent intervertebral disc degeneration in mice

Emanuel J. Novais[1,2,3,4], Victoria A. Tran[1], Shira N. Johnston[1,2], Kayla R. Darris[5,6], Alex J. Roupas[5,6], Garrett A. Sessions[5,6], Irving M. Shapiro[1,2], Brian O. Diekman[5,6] & Makarand V. Risbud[1,2 ✉]

Intervertebral disc degeneration is highly prevalent within the elderly population and is a leading cause of chronic back pain and disability. Due to the link between disc degeneration and senescence, we explored the ability of the Dasatinib and Quercetin drug combination (D + Q) to prevent an age-dependent progression of disc degeneration in mice. We treated C57BL/6 mice beginning at 6, 14, and 18 months of age, and analyzed them at 23 months of age. Interestingly, 6- and 14-month D + Q cohorts show lower incidences of degeneration, and the treatment results in a significant decrease in senescence markers p16[INK4a], p19[ARF], and SASP molecules IL-6 and MMP13. Treatment also preserves cell viability, phenotype, and matrix content. Although transcriptomic analysis shows disc compartment-specific effects of the treatment, cell death and cytokine response pathways are commonly modulated across tissue types. Results suggest that senolytics may provide an attractive strategy to mitigating age-dependent disc degeneration.

[1] Department of Orthopaedic Surgery, Sidney Kimmel Medical College, Thomas Jefferson University, Philadelphia, USA. [2] Graduate Program in Cell Biology and Regenerative Medicine, Jefferson College of Life Sciences, Thomas Jefferson University, Philadelphia, USA. [3] Life and Health Sciences Research Institute (ICVS), School of Medicine, University of Minho, Braga, Portugal. [4] ICVS/3B's—PT Government Associate Laboratory, Braga, Portugal. [5] Thurston Arthritis Research Center, University of North Carolina School of Medicine, Chapel Hill, NC, USA. [6] Department of Biomedical Engineering, University of North Carolina, Chapel Hill, NC, and North Carolina State University, Raleigh, NC, USA. ✉email: makarand.Risbud@jefferson.edu

The prevalence and impact of age-dependent diseases is increasing with increased human lifespan. In a recent global survey of 50 chronic pathological conditions, low back pain (LBP) and neck pain ranked the 1st and 4th top causes of years lived with disability[1]. Although the etiology of LBP is multifactorial, intervertebral disc degeneration is regarded as a major contributor to this pathology[2]. Importantly, aging exacerbates disc degeneration and disease progression, giving rise to an urgent need to understand the underlying mechanisms of disc aging and develop solutions to delay or ameliorate the progression of age-dependent degeneration[2,3].

The intervertebral disc confers flexibility and plays a key role in accommodating the mechanical loads applied to the spinal column. These functional properties are enabled by the interaction of three unique disc compartments: the central nucleus pulposus (NP)—an avascular tissue, rich in aggrecan; the circumferential, ligamentous annulus fibrous (AF)—primarily composed of collagen fibers; and the cartilaginous endplates (CEP) bordering the NP and AF on cranial and caudal surfaces. It is now recognized that abnormal function of any of these compartments can influence degeneration of the others[3,4]. Defining features of disc degeneration include decreased abundance and quality of extracellular matrix (ECM), loss of biomechanical properties, an increase in inflammatory mediators and catabolic processes, and changes in cell phenotype and death[5–7]. Despite widespread prevalence and disease burden, no disease-modifying treatments are currently available for disc degeneration and associated pathologies.

Studies of human tissues and mouse models have shown an increased incidence of senescent cells during intervertebral disc aging and degeneration[8–11]. Senescent cells are broadly characterized by cell cycle arrest, apoptotic-resistance, and the production of catabolic factors known as the senescence-associated secretory phenotype (SASP)[12,13]. Senescence can be induced in response to a variety of stimuli, including telomere attrition, oncogenes, and cell stress (e.g., oxidative, genotoxic, cytokines), which can contribute to SASP activation and senescence transformation[12,13]. Increased expression of cell cycle inhibitors p21, p53, and p16[INK4a] are responsible for maintaining the stable arrest of senescence. Alterations to cell division are accompanied by an increase in cytokines, chemokines, and other SASP proteins. These phenotypic changes culminate in inflammation, fibrosis, loss of regenerative capacity, and ultimately tissue degeneration[13]. Recently, we analyzed $Acan^{CreERT2};p16^{Ink4a}$ conditional knockout mice and found decreased levels of cell death, SASP, and aberrant matrix changes in the discs of aged mice[11]. This finding is supported by reduced oxidative stress and disc degeneration in $Cdkn2a$ germline knockout mice in response to a tail suspension injury model[14]. Additionally, Patil et al.[10] used genetically engineered p16-3MR transgenic mice to demonstrate that systemic clearance of p16[INK4a]-positive cells mitigated age-related disc degeneration by reducing matrix catabolism and senescent cells in the disc compartment. Similarly, Cherif et al.[15,16] showed effective clearing of senescent disc cells and reduction in inflammatory signaling using senolytics RG-7112 and o-Vanillin in ex vivo and in vitro assay systems. While these studies implicate senescent cells in driving disc pathology, it remains to be established if senolytic drugs can slow, cure, or even prevent disc degeneration during physiological aging in vivo.

The concept of senolytic therapy is that apoptosis can be selectively initiated in senescent cells by inhibiting the pro-survival mechanisms upregulated during senescence. Zhu et al.[17] first demonstrated this concept by suppressing the pro-survival BCL-XL and EFNB1 pathways that are highly activated by senescent cells, and other senolytic targets have emerged[18–20]. Of note, the combination of Dasatinib (D)—a Src/tyrosine kinase inhibitor—and Quercetin (Q)—a natural flavonoid that binds to BCL-2 and modulates transcription factors, cell cycle proteins, pro- and anti-apoptotic proteins, growth factors and protein kinases[21,22]—has been used extensively as a senolytic. The D + Q combination has shown beneficial effects in idiopathic pulmonary fibrosis[23], bone loss[24], and improved physical condition and lifespan[25]. In recent human clinical trials, D + Q reduced the number of senescent cells and improved physical performance[26,27]. Cognizant of these results, we investigated the therapeutic potential of D + Q in the context of disc degeneration. Accordingly, we treated wild-type C57BL/6 (BL6) mice beginning at: 6 months—to prevent disc degeneration onset (skeletally mature mice with healthy discs); 14 months—to bar progression of disc degeneration (middle-aged mice with early signs of degeneration); and 18 months—to rescue established disc degeneration (elderly mice with visible degeneration) up to 23 months of age. Our results show that there is a time-dependent effect of the D + Q senolytic cocktail on restoring disc health. This work provides important in vivo evidence that D + Q can, in a non-invasive manner, target senescent cells and mitigate the effects of age-dependent disc degeneration.

## Results

**Dasatinib and Quercetin treatment alleviates age-dependent intervertebral disc degeneration and decreases senescence burden.** Senescence plays an essential role in intervertebral disc aging pathology and disease progression[10,11,14]. To date, no pharmacological approaches to prevent age-related disc degeneration exist. We treated 6-, 14-, and 18-month-old wild-type C57BL/6 (BL6) mice up to 23 months (6–23 M, 14–23 M, 18–23 M), with a weekly injection of a Dasatinib (D) and Quercetin (Q) combination to target senescent cells and prevent and/or treat age-related disc degeneration (Fig. 1a). Histological analysis of lumbar discs from 6–23 M and 14–23 M D + Q cohorts showed better preservation of tissue and cell morphology, with maintenance of NP/AF compartment demarcation and the NP cell band and a lower number of AF clefts relative to Vehicle-treated control animals (Veh) (Fig. 1b and Supplementary Fig. 1a). By contrast, mice in 18–23 M cohort did not show any significant improvements in disc morphology relative to the Veh group (Supplementary Fig. 1b). Histological grades of degeneration for lumbar levels were recorded using a modified Thompson grading system[28,29]. Grade distributions as well as combined average grades showed lower scores of degeneration in the NP and AF compartments of 6–23 M and 14–23 M D + Q cohorts in comparison to respective Veh groups (Fig. 1c, d). Level-by-level analysis of NP degeneration scores showed significantly lower grades in L4-5 and L5-6 in 6–23 M and 14–23 M D + Q mice, respectively, trending toward lower grades at other levels (Fig. 1e and Supplementary Fig. 1c). Similarly, level-by-level analysis of AF degeneration grades showed a significant decrease in scores at L3-4 and L5-6 in the 14–23 M cohort, and grades trended downward in the 6–23 M cohort (Fig. 1e and Supplementary Fig. 1d). Supporting histological observations, the 18–23 M D + Q group presented grade distributions, as well as similar pooled and level-by-level average grades comparable to the Veh group (Supplementary Fig. 1c, d). Together, these results provide strong evidence that D + Q treatment mitigates age-related disc degeneration, a key intervention window exists, and the disease status at the start of the treatment is crucial for therapeutic success.

To evaluate the potential of D + Q in decreasing senescence burden in the disc, we evaluated the abundance of key senescence markers. The 14–23 M cohort was selected for this and subsequent analyses, since these mice showed the most pronounced response to treatment in their disc morphology. Additionally, 14-month-old C57BL/6 mice exhibit early

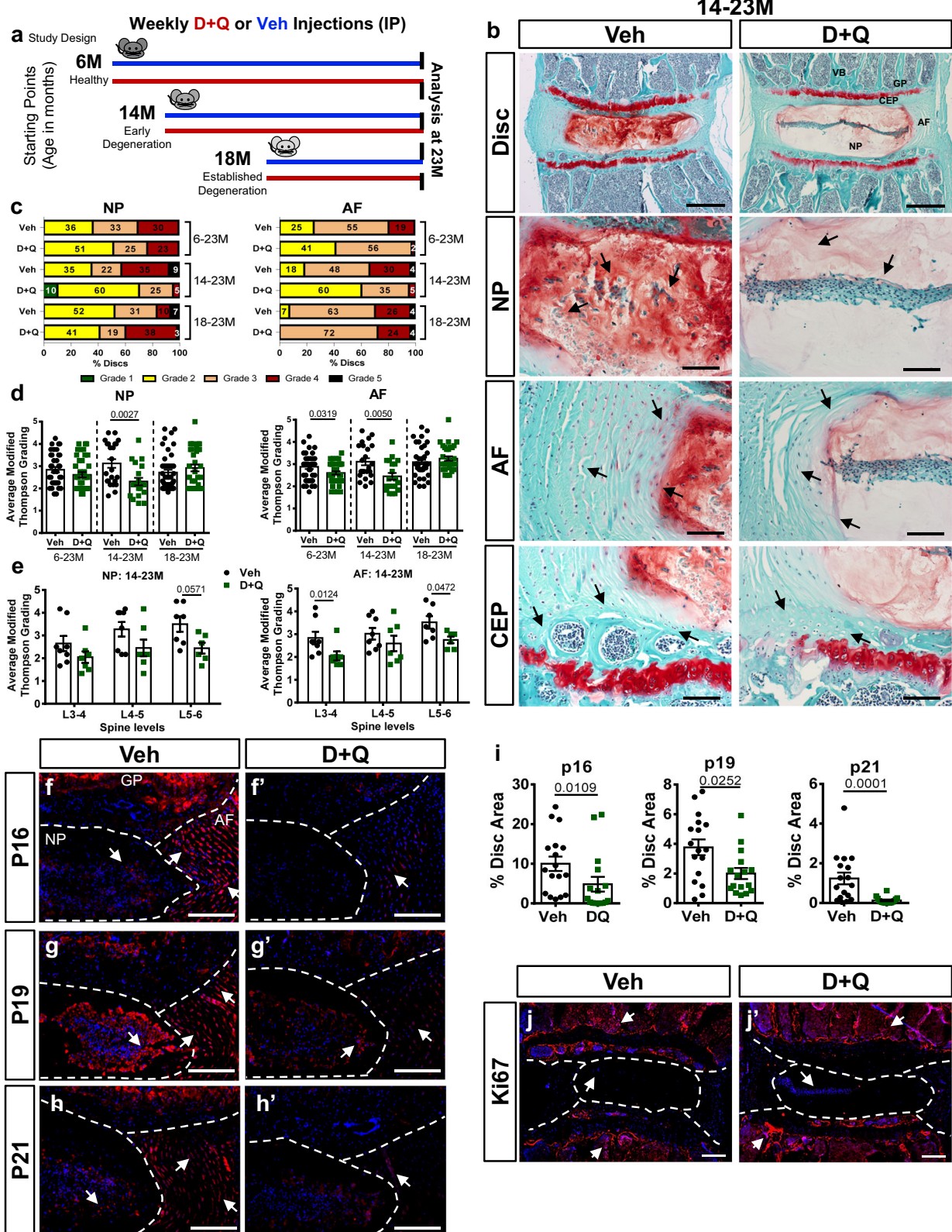

morphological changes associated with age-related disc degeneration[6] and better represent the middle-aged human patient population (ages 40–50 years) with increased LBP prevalence seeking clinical treatments[30]. Interestingly, both p16INK4a and p19ARF, well known markers of senescence in vivo, showed decreased abundance in D + Q-treated discs (Fig. 1f–g', i)[12,13,31–33]. Similarly, levels of p21, a potent cell cycle inhibitor and key downstream mediator of senescence, was also decreased in the D + Q group (Fig. 1h–i)[34]. Moreover, while there were no changes in levels of cell cycle inhibitor RB, levels of pRB were lower in D + Q-treated disc (Supplementary Fig. 2a–d). The decrease in senescence burden by D + Q in disc was not

**Fig. 1 Senolytic drugs Dasatinib and Quercetin alleviate age-dependent intervertebral disc degeneration and decrease abundance of senescence markers. a** Schematic showing study design: peritoneal injections were administered once every week to mice starting at 6 (6 M), 14 (14 M), or 18 (18 M) months of age and ending at 23 (23 M) months of age. Vehicle (1:1 PBS/DMSO), D + Q (5 mg/kg Dasatinib plus 50 mg/kg Quercetin). **b** Histology of the 14–23 M cohort showed preservation of tissue architecture and cell morphology in the NP, AF, and CEP of D + Q-treated mice. 5X and 20X images of 14–23 M show NP, AF, and CEP compartment morphologies. **c, d** Modified Thompson Grading averages and grade distributions showed lower average scores of degeneration in mice that were treated from 6 M and 14 M. Mice in 18–23 M D + Q cohort showed comparable scores the Veh group. **e** Level-by-level analysis of L3-6 in the lumbar spine evidenced decreases in degeneration grades in the NP compartment at L5-6 and in the AF compartment at L3-4 and L5-6. **f–j** Staining and abundance of key markers of senescence p16[INK4a], p19[ARF] and p21, and cell cycle: Ki67 in disc compartments. Two-tailed $t$-test or Mann–Whitney test were used as appropriate for comparing differences between Veh and D + Q groups. 6–23 M Veh ($n = 13$), D + Q ($n = 15$); 14–23 M Veh ($n = 8$), D + Q ($n = 7$); 18–23 M Veh ($n = 11$), D + Q ($n = 9$). Data are represented as mean ± SEM. Scale bar **b** (disc) = 200 µm; Scale bar **b** (NP, AF, and CEP) = 50 µm; Scale bar **f–f'**, **g–g'**, **i–i'**, and **j–j'** = 200 µm. NP: nucleus pulposus; AF: annulus fibrosus; EP: endplate; GP: growth plate, VB: vertebral bone. Source data are provided as a Source Data file.

accompanied by cell proliferation as seen by comparable and low levels of Ki67 staining (Fig. 1j–j'). Of note, the 18–23 M D + Q group, which did not present any alleviation in disc degeneration, showed similar abundance of senescence markers p16[INK4a], p19[ARF], and RB compared to Veh treatment (Supplementary Fig. 2e–g"). Altogether, these results suggest that long-term treatment of D + Q successfully targets and reduces the number of senescent cells in the intervertebral disc.

**D + Q treatment mitigated age-dependent progression of SASP and preserved NP cell phenotype and viability.** SASP is known to promote tissue inflammation, catabolism, and fibrosis, which contribute to the progression of degenerative pathologies[13]. To study the age-dependent progression of SASP markers in the disc and whether they are responsive to D + Q, we evaluated the abundance of IL-1β, IL-6, and MMP13 (Fig. 2a–c"), well-established SASP markers in different tissues, including the intervertebral disc[10,11,18,35]. In addition to the Veh and D + Q groups (14–23 M cohort), we analyzed 1-year-old BL6 (1 y Ctr) mice as a reference for the healthy adult state and to understand the effect of aging on measured parameters. Quantitative immunohistochemical analyses showed reductions in IL-6 and MMP13 in both D + Q and 1 y Ctr relative to the Veh group (Fig. 2b, c"). In contrast, higher levels of IL-1β were seen in D + Q and 1 y Ctr groups (Fig. 2a–a"). To investigate the effects of D + Q on NP cell phenotype and disc cell viability, CA3 and GLUT1 abundance were assessed[36], and TUNEL staining was performed. While NP cells from D + Q-treated and 1 y Ctr mice robustly expressed CA3 and GLUT1, the Veh group showed decreased abundance of these markers (Fig. 2d, e"). Moreover, discs of D + Q-treated mice showed higher cellularity and lower percentages of TUNEL-positive cells (Fig. 2f–f"). Interestingly, discs from 1 y Ctr mice had comparable cell counts and percentages of TUNEL-positive cells with the D + Q cohort (Fig. 2f–f"), suggesting a cytoprotective effect of the treatment on cells. Together, these results suggest that D + Q treatment decreases SASP and preserves NP cell phenotype and viability during aging.

**D + Q treatment preserved healthy ECM and limited Aggrecan degradation during aging.** Integrity of the ECM is vital to proper disc function. Importantly, aging promotes ECM catabolism, leading to decreased levels of proteoglycans and collagens[37] and compromised tissue function[38]. COL1, COL2, and COMP abundances were lower in discs from the Veh group (14–23 M cohort) compared to the 1 y Ctr (Fig. 3a–c"). Interestingly, while COL2 levels did not change in D + Q group, COL1 and COMP levels were higher compared to the Veh group (Fig. 3a–c"). Moreover, COLX, a collagen type highly expressed during disc degeneration[7], was present at lower levels in both 1 y Ctr and

D + Q groups compared to the Veh group (Fig. 3d–d"). Chondroitin sulfate is a functional constituent of aggrecan, a proteoglycan responsible for water binding and viscoelastic properties of the disc[39]. Despite comparable abundance of CS between 1 y Ctr and Veh groups, D + Q-treated mice showed higher abundance of CS compared to the Veh group (Fig. 3e–e"). Additionally, abundance of ARGxx, a degradation product generated following aggrecan cleavage, was significantly lower in discs of D + Q-treated mice compared to the Veh-treated group (Fig. 3f–f"). However, there were no significant differences in overall abundance of ARGxx levels between the discs of 1 y Ctr and Veh animals, suggesting that aggrecan turnover likely begins earlier than thought (Fig. 3f–f'). These results suggest that D + Q treatment helps mitigate the loss of disc ECM with aging.

**D + Q treatment reduces nucleus pulposus fibrosis during aging.** Fibrosis is one of the main degenerative disc phenotypes promoted by aging[28]. Collagen content, as measured by imaging FTIR spectroscopy, showed lower levels in the NP compartment of D + Q-treated mice when compared to Veh-treated animals (Fig. 4a, b). Similarly, Picrosirius Red staining with polarized microscopy showed that only 12% of D + Q discs possessed distinct collagen fibers within the NP compartment. The vehicle group had collagen fibers in 50% of discs, whereas there were no quantifiable collagen fibers in the NP from 1-year controls (Fig. 4c, d). Assessment of collagen fiber thickness in the AF compartment showed that D + Q group presented the lowest percentage of thick fibers (red) and an overall increase in thin fibers (green) as compared to Veh and 1-year Ctr mice (Fig. 4d, e). We also assessed the collagen quality and integrity by treating disc sections with fluorescently labeled collagen hybridizing peptide (CHP), which selectively binds to denatured collagen[40]. There were no differences in CHP binding among the groups (Fig. 4f).

**D + Q treatment resulted in distinct transcriptomic modulation in AF and NP compartments.** Microarray analysis was performed on AF and NP tissues from discs of 14–23 M Veh- and D + Q-treated mice. Baseline differences between AF and NP compartments were first assessed by analyzing the differentially expressed genes (DEGs) between AF-Veh and NP-Veh with a cutoff of $p < 0.05$ (Fig. 5a–c and Supplementary Fig. 3). The transcriptomic profiles of each tissue clustered distinctly, as demonstrated by principal component analysis (Fig. 5a), and a similar number of the 10,283 total DEGs were upregulated in each tissue (Supplementary Fig. 3a). Expectedly, NP tissue showed higher levels of *Krt19*, *Slc2a1*, and *Car3*, well-established markers of the NP compartment; the AF presented higher levels of *Col1a2*, *Comp*, and *Ibsp*, known hallmarks of the AF compartment (Fig. 5b)[41,42]. Importantly, each tissue presented with unique regulation of genes related to the ECM, focal adhesions,

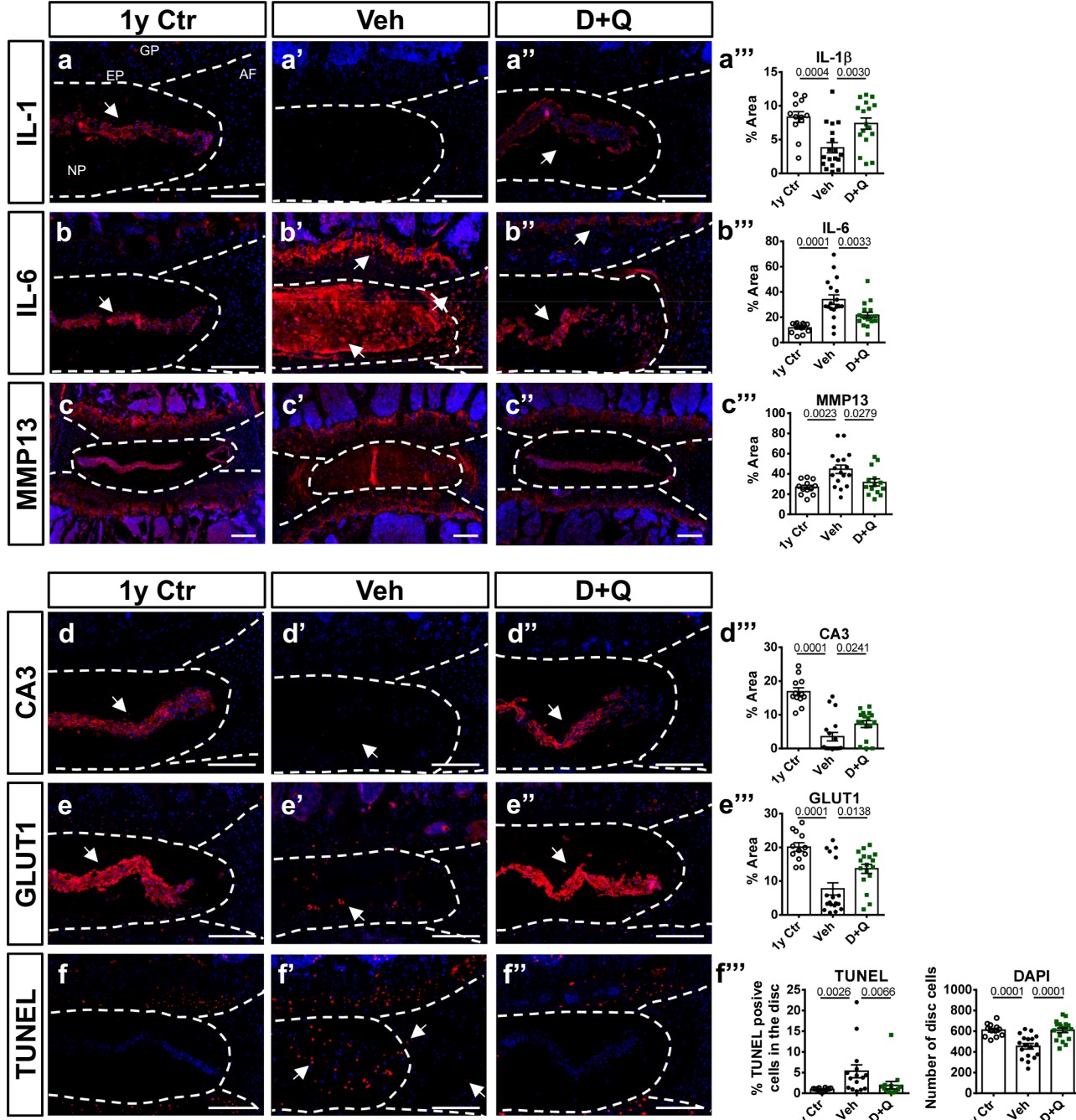

**Fig. 2 D + Q treatment prevents age-dependent SASP progression and promotes NP cell phenotype maintenance and disc cell viability. a–c'''** Immunohistological staining and staining area quantifications of SASP markers IL-1, IL-6, and MMP13 in the disc compartments of 1-year-old healthy Ctr (1 y Ctr), 14–23 M Veh-, and 14–23 M D + Q-treated mice. **d–e'''** Quantitative immunostaining of NP cell phenotypic markers CA3 and GLUT1. **f–f'''** Cell death assessment in the disc compartments by TUNEL staining. Two-tailed ANOVA or Kruskal–Wallis tests were used as appropriate; $n = 6$ mice/group, 2–3 levels per mouse. Data are represented as mean ± SEM. Scale bar = 200 μm. Source data are provided as a Source Data file.

and phenotype regulation in 23-month-old mice (Supplementary Fig. 3b, c). Additionally, related to regulation of the cell cycle, higher levels of *Cdkn2d*, *Ccnd2*, *Pcna*, and *E2f2* were seen in the AF, while the NP compartment showed elevated levels of *Cdkn1c*, *Cdkn1b*, *Cdkn2b*, *Ccnd1*, and *Atm* (Fig. 5c). This data suggests that AF and NP compartments present distinct transcriptomic profiles at 23-months, with divergent regulation of critical biological pathways, including the cell cycle.

To explore the biological pathways targeted by D + Q treatment in each disc compartment, AF D + Q and Veh groups

were compared, identifying 1,646 DEGs. Of these, 65% were upregulated, and 35% were downregulated in the D + Q group; $p < 0.05$ (Fig. 5d and Supplementary Fig. 4a, b). To understand the biological impact of these DEGs, gene enrichment analysis of these up- and downregulated DEGs groups was performed using the Overrepresentation Test in PANTHER[43]. The DEGs upregulated in the AF of D + Q-treated mice showed enrichment of G protein-coupled receptor (GPCR) signaling, response to stress, mitotic cell cycle, cellular response to DNA damage stimulus, cell adhesion, and DNA repair pathways (Fig. 5e). We

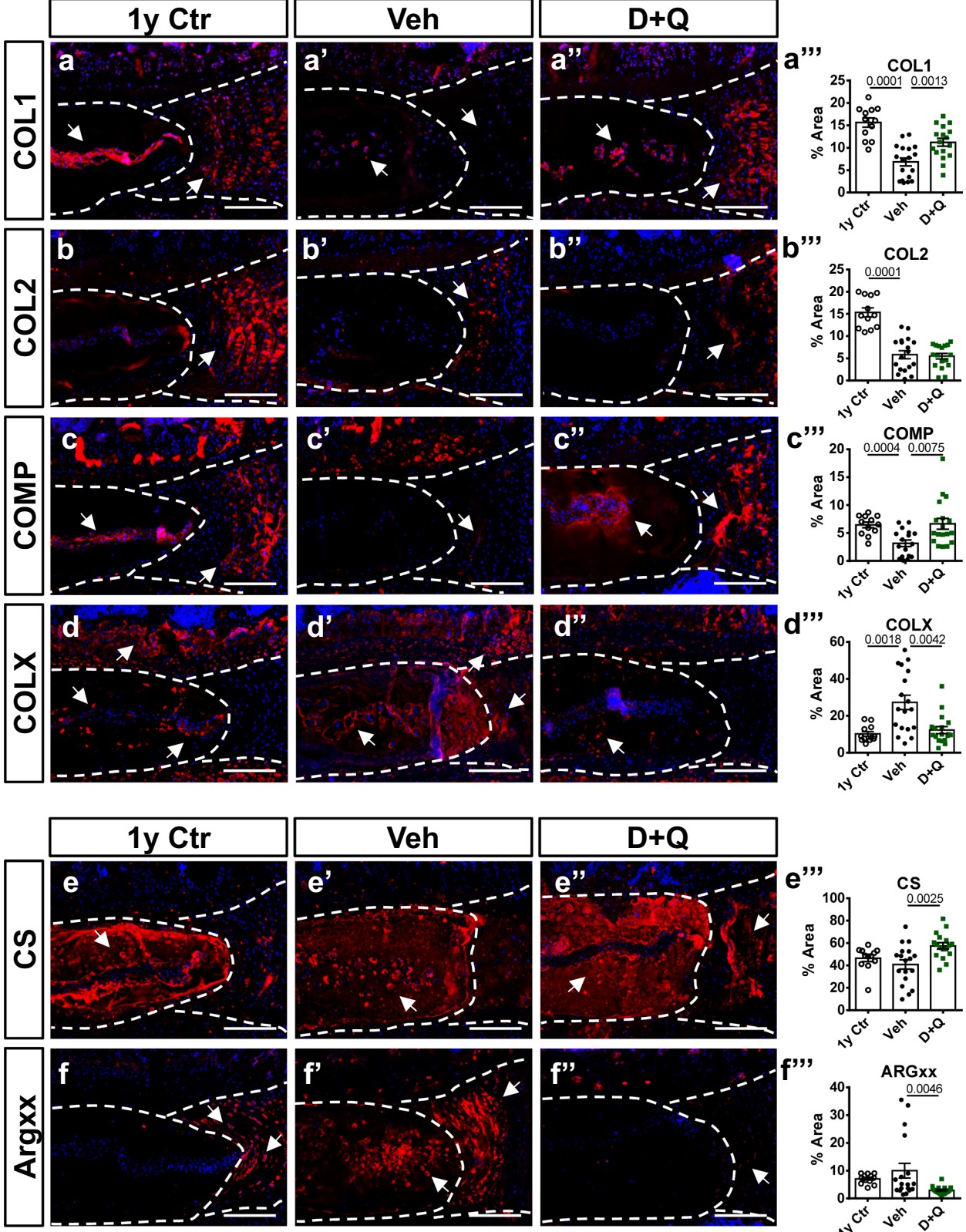

**Fig. 3 D + Q treatment preserved healthy extracellular matrix composition with a decrease in Aggrecan degradation during aging. a–f‴** Staining and abundance in the disc compartment of essential extracellular matrix proteins in 1 y Ctr, 14–23 M Veh, and 14–23 M D + Q groups: Collagen I, Collagen II, COMP, Collagen X, Chondroitin sulfate, and ARGXX neoepitope. Two-tailed ANOVA or Kruskal–Wallis tests were used as appropriate; n = 6 mice/group, 2–3 levels per mouse were analyzed. Data are represented as mean ± SEM. Scale bar **a–f**″ = 200 μm. Source data are provided as a Source Data file.

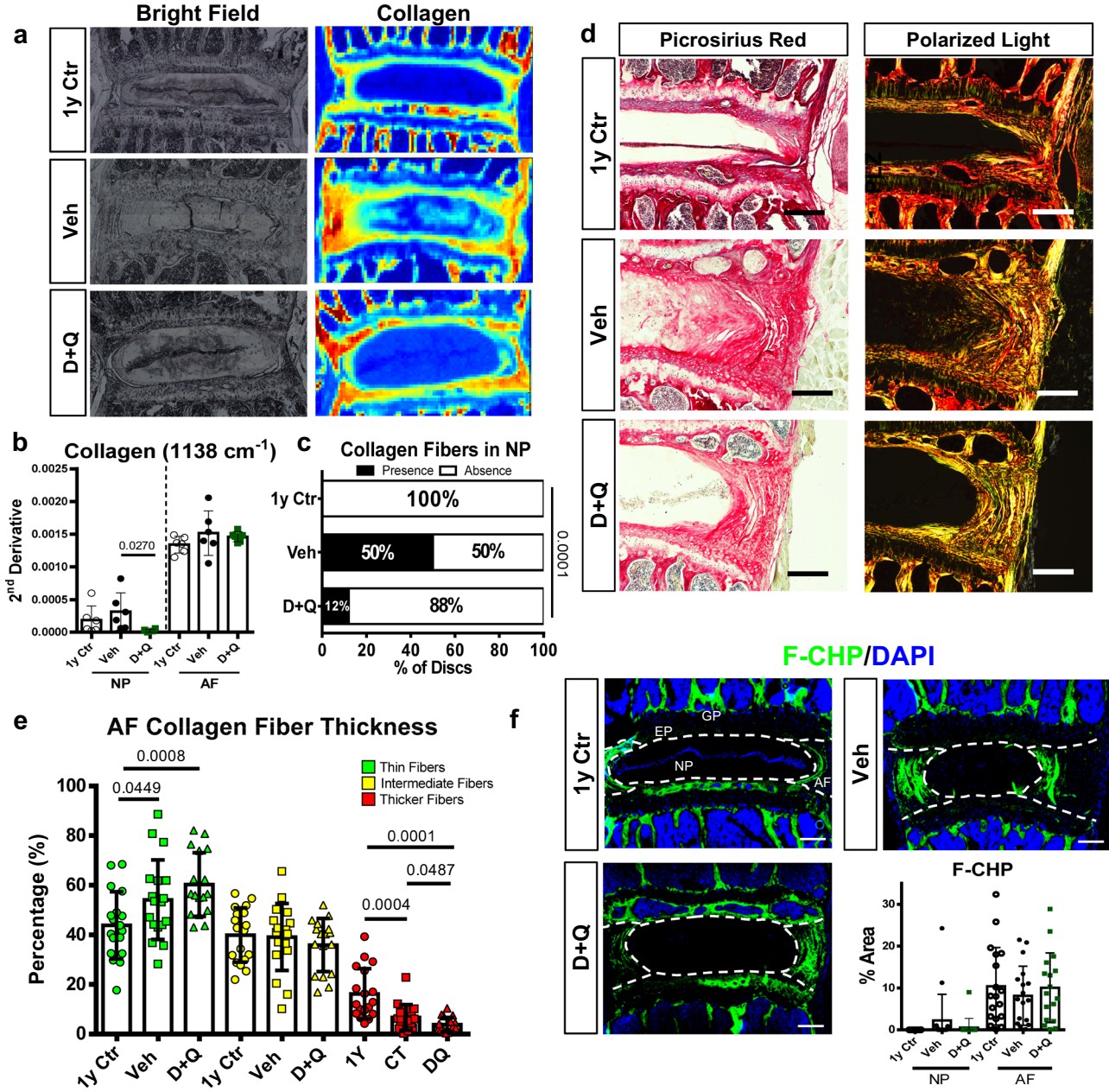

**Fig. 4 D + Q treatment results in lower incidence of NP fibrosis. a** Bright field and chemical map of mean second-derivative peaks for collagen (1338 cm$^{-1}$). Red: high relative absorbance; yellow: intermediate relative absorbance; blue: low relative absorbance. **b** Quantification of mean second-derivative peaks for collagen (1338 cm$^{-1}$). **c** Quantification of the percentage of presence/absence of collagen fibers within the NP compartment. **d** 1-year Ctr, Veh and D + Q lumbar discs showing collagen fiber organization under bright field and polarized light (scale bar = 100 μm). **e** Quantification of collagen fiber thickness distribution and proportion in the AF. **f** Denatured collagen, measured by a collagen hybridization peptide (CHP) biding assay (scale bar = 200 μm). Two-tailed ANOVA or Kruskal–Wallis tests were used as appropriate; $n = 6$ mice/group, 2–3 levels per mouse were analyzed. Data are represented as mean ± SEM. Source data are provided as a Source Data file.

found several AF DEGs associated with negative regulation of cell death: *Hhip*, *C7*, *Pcp4*, *Cntfr*, *Il2*, *Serpinb13*, *Card13*, and *Il7*; mitotic cell cycle: *Slxl1*, *Morc2b*, *Stard9*, *Piwil2*, *Rab11*, *fip3*, *Trim36*, *Irf6*, *Myog*, and *Tppp*; and DNA repair and response to stress: *Cd36*, *Mid1*, *Morc2b*, *Vsig4*, *Ccr7*, *Spo11*, *Gpx2*, *Serpinb3a*, *Ccl1*, *Mink1*, *Il2*, *Il21*, and *Cxcl5* (Supplementary Fig. 4c). On the other hand, downregulated AF DEGs were enriched for ribosome biogenesis, mitochondria organization, cellular respiration, response to cytokines, ATP metabolic process, regulation of cell death, as well as GPCR signaling (Fig. 5f). Noteworthy DEGs were identified in relation to cellular respiration: *Atp5d*, *Atp5f1*,

*Atp5o*, *Atp5g1*, *Ndufs8*, *Nduf7*, *Ndufc2*, *Ndufb5*, *Ndufb8*, *Sdhd*, *Dld*, *Cox6a1*, and *Sdhaf2*; ribosome biogenesis: *Upt3*, *Rpl38*, *Rps16*, *Ddx52*, *Rps15*, *Rps17*, *Rps21*, *Rps5*, *Rpl38*, *Rpl6*, *Mrpl20*, and *Fastkd2*; and response to cytokines: *Myc*, *Socs3*, *Nfil3*, *Junb*, *Mcl1*, *Irf1*, *Klf6*, *Hsp90aa1*, *Vamp8*, *Icam1*, *Cebpb*, *Tnfrsf1a*, *Igbp1*, *Il6ra*, *Hspa5*, *Mapk3*, *Cd14*, and *Irak2* (Supplementary Fig. 4d).

Analysis of DEGs comparing NP D + Q and Veh groups identified 964 DEGs, of which 54% were upregulated, and 46% downregulated, $p < 0.05$ (Fig. 5g and Supplementary Fig. 4e, f). Importantly, DEGs downregulated in the D + Q group showed enrichment in the regulation of DNA-templated transcription in

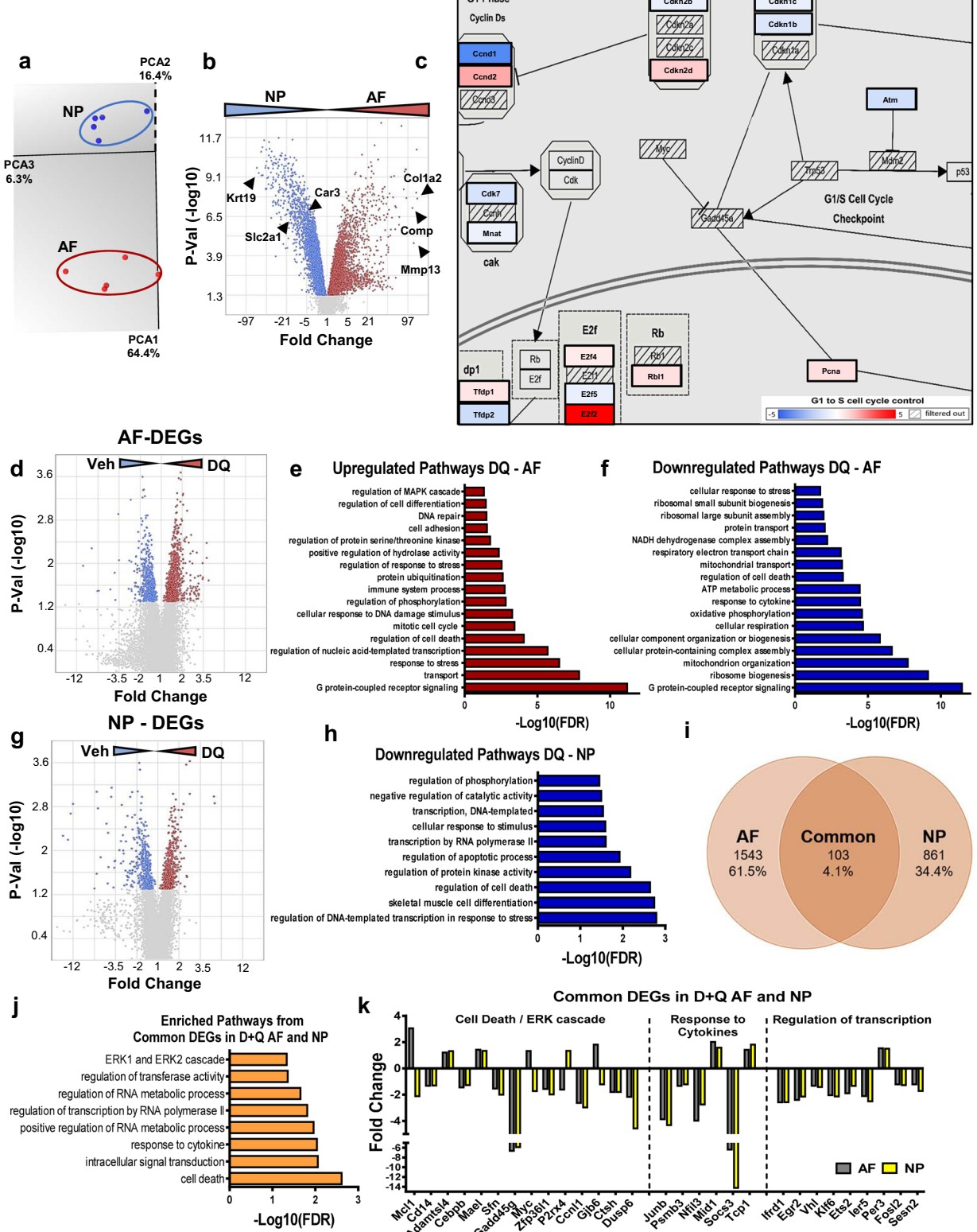

response to stress, skeletal muscle cell differentiation, regulation of cell death, protein kinase activity, apoptotic process, transcription by RNA polymerase II, and regulation of phosphorylation pathways (Fig. 5h). Noteworthy DEGs were identified in relation to skeletal muscle cell differentiation: *Atf3*, *Nr4a1*, *Btg2*, *Maff*, *Egr1*, *Sap30*, *Scx*, and *Asb2*; transcription by RNA polymerase II: *Fosb*, *Fos*, *Junb*, *Jun*, *Nfil3*, *Egr2*, *Atf4*, and *Jund*; regulation of protein kinase activity: *Errfi1*, *Gadd45g*, *Dusp6*, *Egr1*, *Cyr61*,

*Dusp1*, *Dusp8*, *Taf7*, *Hspb1*, *Efna1*, *Cdkn2d*, *Sesn2*, *Pdgfc*, *Mapk3k2*, and *Gadd45b*; and regulation of cell death: *Pim1*, *Cyr61*, *Klf4*, *Serpine1*, *Id3*, *Mcl1*, *Rhob*, *Pim3*, *Efna1*, *Axl*, *Myc*, *Itga5*, *Eif2b5*, *Cdh1*, *Aven*, and *Bdnf* (Supplementary Fig. 4g). Interestingly, DEGs upregulated in the D + Q-treated NP showed enrichment for zinc finger transcription factor, defense/immunity protein, and immunoglobulin related molecular classes (Supplementary Fig. 4h, i). Some of these genes include *Ersr1, Ppox,*

**Fig. 5 D + Q treatment promoted transcriptomic modulation of cell death, response to cytokines, regulation of RNA polymerase II, and the ERK1/ERK2 cascade in AF and NP compartments. a** Transcriptomic profiles of AF ($n = 5$) and NP ($n = 5$) tissues from 14–23 M Veh mice clustered distinctly with principal component analysis. **b** Volcano Plot, showing up- and downregulated DEGs from the 23 M AF vs. NP comparison used for GO Process enrichment analysis, $p$-value ≤ 0.05. **c** Schematic summarizing the DEGs between the AF and NP at 23 M related to cell cycle regulation. **d** Volcano Plot, showing up- and downregulated DEGs from the D + Q AF vs. Veh AF comparison, $p ≤ 0.05$. **e** Representative GO processes of upregulated genes from D + Q AF vs. Veh AF. **f** Representative GO processes of downregulated genes from D + Q AF vs. Veh AF. **g** Volcano Plot, showing up- and downregulated DEGs from the D + Q NP vs. Veh NP comparison, $p$-value ≤ 0.05. **h** Representative GO processes of downregulated genes from D + Q NP vs. Veh NP. **i** Venn Diagram of common DEGs from D + Q AF vs. Veh AF and D + Q NP vs. Veh NP, $p$-value ≤ 0.05. **j** Representative GO processes of common DEGs from panel **I**. **k** Representative DEGs from selected GO processes from panel **j**. $p$-values were calculated using eBayes ANOVA Method (unadjusted). GO analysis was performed using the PANTHER Overrepresentation Test with GO Ontology database annotations and a binomial statistical test with FDR ≤ 0.05. Source data are provided as a Source Data file.

_Polg2, Hoxd10, Slc4a1, Xlr4b, Ppp1r10, Fbxw26, Tlr7, Snapc3, Lrif1, Zkscan7, Tet2, Dmrta1, Zfp760, Ermap, Ogt, Ankzf1, Lipf, Apol11b, Ddc, Il1rpl1, Adamts20, Eaf2_, and _Xlr4a_.

To further explore the pathways commonly modulated in the AF and NP by D + Q treatment, DEGs identified in both the AF and NP D + Q vs. Veh. analyses were investigated. From this analysis, 103 common DEGs emerged (Fig. 5i). Analysis of these DEGs showed enrichment for cell death, response to cytokines, regulation of RNA polymerase II, and ERK1/ERK2 cascade pathways (Fig. 5j). The majority of DEGs related to cell death/ERK cascade were downregulated in the D + Q groups and included _Cd14, Cebpb, Sfn, Gadd45g, Zfp36l, Ccnl1, Ctsh_, and _Dusp6_. Common DEGs related to the response to cytokines included _Junb, Psmb3, Nfil3_, and _Socs_, whereas _Ifrd1, Egr2, Vhl, Klf6, Ets2, Ier5, Fosl2_, and _Sesn2_ were linked to regulation of transcription (Fig. 5k). Overall, these results suggest that AF and NP compartments in old mice possess distinct baseline transcriptomic profiles, and D + Q treatment results in modulation of both unique and common pathways in each tissue.

Finally, to study the effect of D + Q treatment on aging-associated DEGs in the NP, we compared the DEGs modulated by D + Q in 23 M mice with the transcriptomic signature of aging in C576BL/6 mice (DEGs obtained from 23 M vs. 6 M naive BL6 NP: GSE134955, $p < 0.05$)[28]. The transcriptomic profiles of 23 M and 6 M BL6 groups clustered distinctly, as demonstrated by principal component and hierarchical clustering analyses (Fig. 6a, b). Comparison of aging (23 M vs. 6 M) and D + Q modulated DEGs showed 166 common genes (Fig. 6c). Interestingly, the enrichment analysis of these common DEGs showed association with negative regulation of apoptotic process, regulation of kinase activity, regulation of cell differentiation, regulation of cell population proliferation, cellular response to oxygen-containing compound, and transcription by RNA polymerase II (Fig. 6c). Noteworthy, the DEGs related to negative regulation of protein kinase activity were _Gadd45g, Dusp6, Dusp1, Taf7, Gprc5a, Prkar1b, Gadd45a_, and _Deptor_. Common DEGs related to negative regulation of apoptotic process included _Socs3, Jun, Klf4, Mcl1 Ctsh, Axl, Myc, Atf5_ and _Adamts20_. Whereas _Atf3, Nr4a1, Fos, Btg2, Maff, Egr1_ and _Egr2_ were linked to skeletal muscle cell differentiation (Fig. 6d). We then focused our analysis on the group of common genes that were modulated in different directions by aging and D + Q treatment (Fig. 6e). From this analysis we found 27 genes upregulated during NP aging but downregulated in the D + Q group, namely: _Mbp, Axl, Cdh19, Pdgfc, Deptor, Ctsh, Cpe, Jup, Ar, Ica1l, Cox5a, Snd1, Tubb2b, Bbs4_, and _Psmd4_ (Fig. 6e, f). Likewise, 19 genes including _Dclk1, Adamts20, E2f7, Fam19a1, Erc2, Adam4, Rnpc3, Cyp2c66, Tet2, Rassf7, Dsg3_, and _Eddm3b_ were downregulated during aging but were upregulated by D + Q (Fig. 6e, f). Overall, these results suggest that D + Q modulates critical genes associated with aging in the NP, suggesting its potential to slow intervertebral disc degeneration during aging.

**D + Q treatment mitigated an age-associated increase in circulatory levels of proinflammatory cytokines and Th17-related proteins.** To explore the impact of aging and D + Q treatment on the status of systemic inflammation, we performed multiplex analysis of several inflammatory molecules from plasma; Young (6 months) vs. Old BL6 (23 months) and D + Q vs. Veh-14–23 M cohorts were analyzed at the time of sacrifice. Data from 14–23 M treatment are shown in Fig. 7, with data from the 6–23 M and 18–23 M shown in Supplementary Fig. 5. Aged mice showed increases in: proinflammatory proteins IFN-γ, IL-2, IL-6, and TNF-α; cytokines IL-27p28/IL-30, and MCP-1; and Th17 inflammatory mediators IL-16, IL-17A, IL-17C, IL-21, and MIP-3α (Fig. 7a–c). An increase in anti-inflammatory cytokine IL-10 was also noted with aging (Fig. 7a). Interestingly, there was a significant decrease in plasma levels of IL-1β in all three D + Q-treated cohorts. Moreover, the 6–23 M D + Q group showed diminished levels of IL-6 and TNF-α, and the 18–23 M D + Q cohort showed decreased levels of IL-2 relative to the Veh group (Fig. 7a' and Supplementary Fig. 5a–a'). Levels of IL-15, IL-17A/F, IL-27p28/IL-30, IL-33, IP-10, MIP-2, MIP-1α/CCL3, and MCP-1 did not show any significant modulation between Veh and D + Q groups for all 3 cohorts (Fig. 7b' and Supplementary Fig. 5b–b'). Noteworthy, while the 18–23 M cohort did not present any change in Th17-related molecules, the 14–23 M D + Q group showed decreased levels in IL-16, IL-17E/IL-25, IL-21, IL-22, and IL-31 (Fig. 7c' and Supplementary Fig. 5c'). Similarly, the 6–23 M D + Q group showed decreased levels of IL-17C, IL-22, and IL-31 (Supplementary Fig. 5b). These results suggest that despite increased systemic inflammation with aging, D + Q treatment resulted in an overall suppression of proinflammatory and Th17 mediators when treatment was initiated at earlier time-points.

**Prolonged D + Q treatment was well tolerated and beneficial effects varied between musculoskeletal tissue types.** Previous studies have shown a positive effect of D + Q treatment on different tissues. However, the optimal starting point and duration of treatment in age-related-pathologies is not well explored[24,25,44]. We therefore analyzed survival percentage, change in body weight, vertebral bone properties, and osteoarthritis status in the knee joints in 23-month-old mice from three treatment cohorts. Mice in D + Q-treated groups showed comparable survival rates to the Veh-treated animals and were in accordance with strain life expectancy, as established by Jackson Laboratory's lifespan aging studies (Fig. 8a and Supplementary Fig. 6a, d). D + Q-treated mice, with the exception of a trend in female mice from the 6–23 M cohort, showed comparable weight progression during aging when compared to Veh controls (Fig. 8b, c and Supplementary Fig. 6b, c, e, f). As part of the frailty index evaluation, the grip strength of the 6–23 M cohort was measured. Interestingly, D + Q mice exerted greater force than the Veh group (Supplementary Fig. 6g). We also assessed whether D + Q treatment influenced disc height and disc height index (DHI). While none of the 3 cohorts showed significant differences

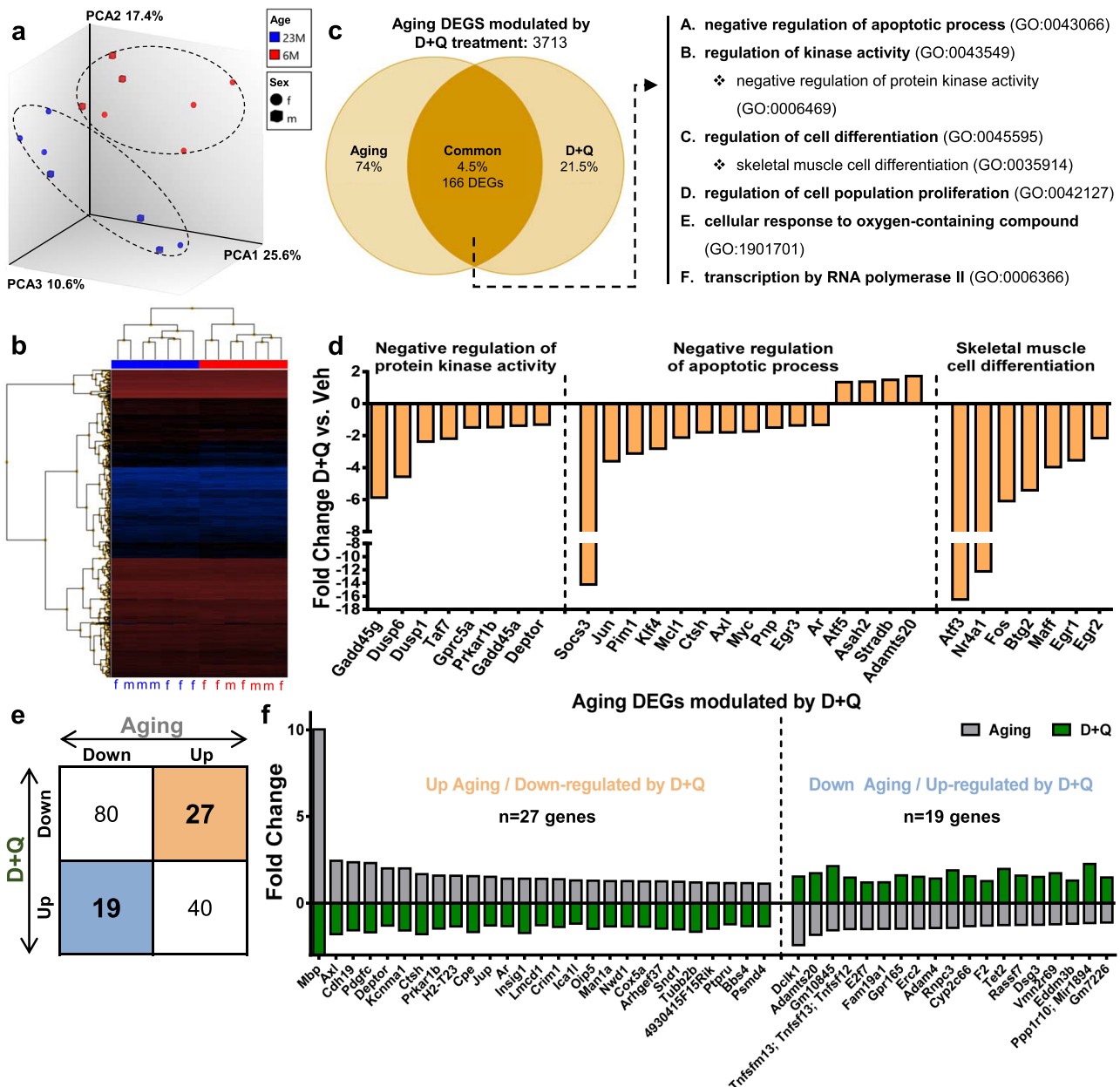

**Fig. 6 D + Q modulates aging-associated DEGs related to negative regulation of apoptotic process, regulation of kinase activity and regulation of cell differentiation pathways. a**, **b** PCA and hierarchical clustering of transcriptomic signature from 23 M vs. 6 M BL6 NP of deposited data in GSE134955, $p < 0.05$[28]. **c** Venn diagram from intersection analysis of aging and D + Q modulated DEGs and the representative GO processes associated with the 166 common DEGs. **d** Representative common DEGs from selected GO processes from panel C. **e** Schematic of common DEGs showing directionality of change in each experiment study (Aging vs. D + Q). **f** Representative DEGs with opposite modulation between aging and D + Q selected from panel **e**. $p$-values were calculated using eBayes ANOVA Method (unadjusted). GO analysis was performed using the PANTHER Overrepresentation Test with GO Ontology database annotations and a binomial statistical test with FDR ≤ 0.05. Source data are provided as a Source Data file.

in DHI, there was a trend towards increased disc height in 6–23 M D + Q-treated mice, compared to Veh-treated mice (Fig. 8d and Supplementary Fig. 7a–c). Mice from the 18–23 M D + Q cohort showed an increase in trabecular and cortical bone thickness, as well as cortical bone area in males (Supplementary Figs. 8 and 9). However, changes in trabecular and cortical bone parameters were not observed between Veh- and D + Q-treated mice in 6–23 M and 14–23 M cohorts (Fig. 8e–f and Supplementary Figs. 8 and 9). Similarly, bone mineral density (BMD) analysis revealed no significant changes between Veh and D + Q groups across the three cohorts, and as expected, female but not

male mice showed an age-dependent (12 M vs. 23 M) decrease in BMD (Supplementary Fig. 10). Additionally, the extent to which D + Q mitigated the development of hindlimb OA was investigated. To provide a measure of cartilage degeneration, the Osteoarthritis Research Society International (OARSI) score[45] was summed across the four quadrants of the joint. The distribution of histological scores was similar between the Veh and D + Q treatments in the 14–23 M cohort (Fig. 8g). The wide distribution in scores was at least partly driven by sex-based differences, but there was no trend towards improved scores in either males or females with D + Q treatment in any of the

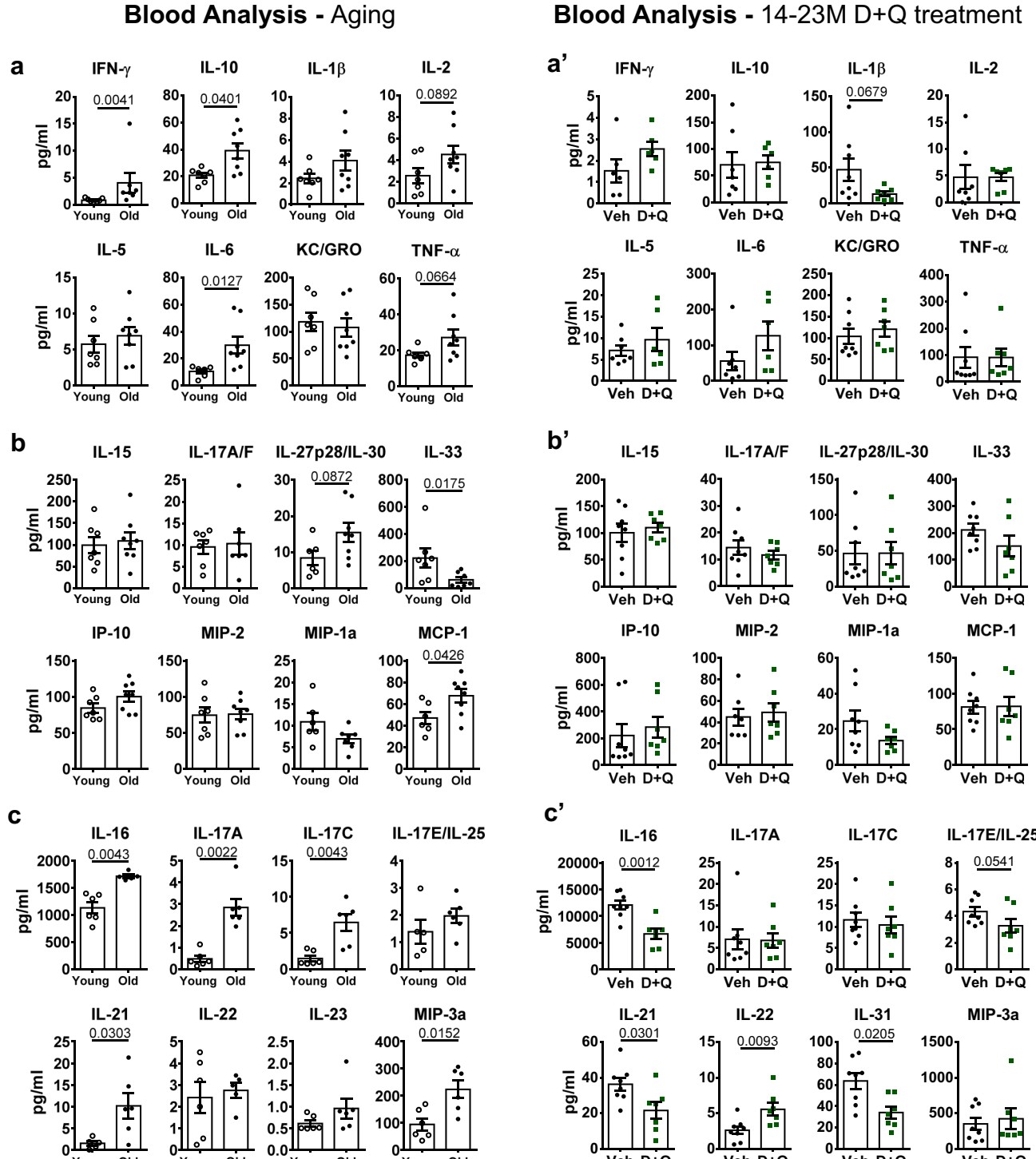

**Fig. 7 D + Q treatment prevented the age-associated systemic increase in proinflammatory molecules, cytokines, and Th17-related proteins.** Multiplex analysis of the main (**a–a'**) proinflammatory molecules, (**b–b'**) cytokines, and (**c–c'**) Th17-related proteins in plasma of young (6–12 M, n = 7), old (23 M, n = 8), Veh-14–23 M (n = 8), and D + Q-14–23M (n = 7) groups. Two-tailed t-test or Mann–Whitney test were used as appropriate. Data are represented as mean ± SEM. Source data are provided as a Source Data file.

treatment duration cohorts (Supplementary Fig. 11a, b). Noteworthy, unlike in the intervertebral disc, immunohistochemistry analyses of senescence markers p16^INK4a, p19^ARF, and p21 in the knee joints did not show any differences between 14 and 23 M Veh- and DQ-treated mice (Supplementary Fig. 11c–e). Taken together, these results suggest that the prolonged D + Q treatment is well tolerated by mice. Interestingly, the treatment

showed differential effects on progression of aging pathologies depending on the skeletal tissue type (i.e., disc, bone, or cartilage) and starting point of treatment.

## Discussion
Despite the staggering medical and societal costs of treating the multitude of pathologies associated with intervertebral disc

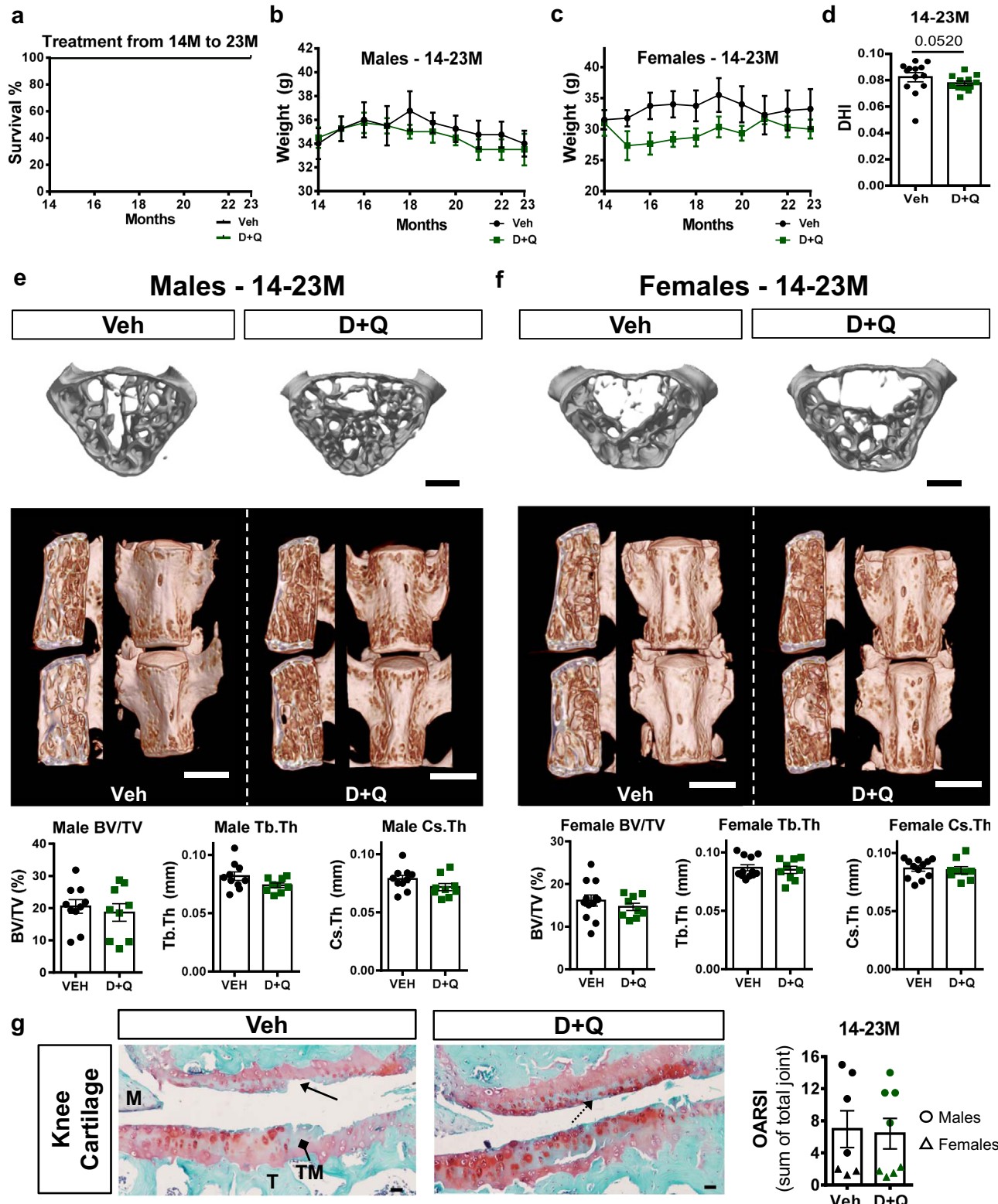

degeneration, there are few available clinical interventions, with surgeries for symptomatic relief having suboptimal outcomes[46]. Recent studies have suggested that senescence is an important contributor to the progression of disc degeneration[10,11,14]. In this study, we evaluated the therapeutic potential of the senolytic drug combination Dasatinib and Quercetin, which has shown promising results in other tissues, to treat progression of age-

dependent disc degeneration. Additionally, we determined whether disc health outcomes vary depending on treatment starting points, as well as the systemic impact of long-term D + Q treatment. Our results show that the D + Q combination could target senescence in the mouse disc, and these results provide proof of principle that senolytics may be useful in mitigating age-dependent disc degeneration by decreasing local senescence

**Fig. 8 Long-term D + Q treatment was well tolerated by mice. a** Survival curve from the 14–23 M Veh and. D + Q cohorts. **b** Weight progression in males from 14–23 M Veh ($n = 4$) and D + Q ($n = 4$) cohorts. **c** Weight progression in females from 14–23 M Veh ($n = 4$) and D + Q ($n = 3$) cohorts. **d** Disc height index comparison between 14 and 23 M Veh and D + Q groups. **e** Representative lumbar spine μCT pictures and bone properties:BV/TV, trabecular thickness, and cortical bone thickness of Veh ($n = 4$) and D + Q ($n = 4$) males from the 14–23 M cohort, L4-L6/mouse. **f** Representative lumbar spine μCT pictures and bone parameter analysis of BV/TV, trabecular thickness, and cortical bone thickness of Veh ($n = 4$) and D + Q ($n = 3$) females from the 14–23 M cohort, L4-L6/mouse. Scale bar **e**, **f** = 500 μm. *t*-test or Mann–Whitney test was used as appropriate. **g** Representative hindlimb histology from the 14–23 M cohort as assessed by Safranin-O/Fast Green/Hematoxylin. Imaged from the lateral side of the joint. Anatomical and histological features of the joint are denoted: Femur (F), Tibia (T), meniscus (M), tidemark (TM with diamond arrow), Saf-O loss (dotted arrow), and vertical clefts of cartilage loss (solid arrow). OARSI grading (0–6) summed across 4 quadrants of joint from 14–23 M Veh ($n = 7$) and D + Q ($n = 8$) cohorts. Males denoted by circles, females by triangles. Scale bar **e**, **f** = 500 μm, G = 20 μm. Two-tailed *t*-test or Mann–Whitney test was used as appropriate. Data are represented as mean ± SEM. Source data are provided as a Source Data file.

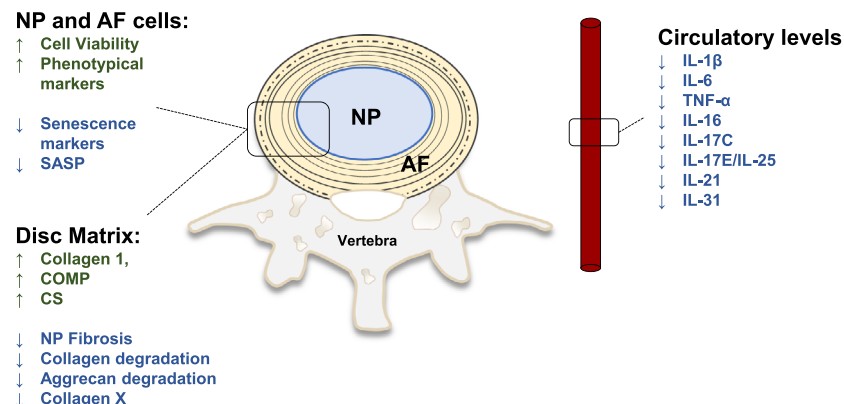

**Fig. 9 Senolytics ameliorate age-related disc degeneration. a** Schematic summarizing the overall beneficial effects of D + Q contributing to alleviation of intervertebral disc degeneration during aging.

status, fibrosis and matrix degradation, while promoting cell viability, healthy matrix deposition and lower levels of systemic inflammation (Fig. 9a).

Zhu et al.[17] first discussed D + Q treatment and introduced a new class of drugs, senolytics, to selectively target senescent cells by interfering with their pro-survival pathways. Among senolytics, the D + Q combination has shown high efficiency in targeting senescent cells[23] and alleviating age-related pathologies[24,25,47]. Noteworthy, previous studies have shown an increase in senescence during disc degeneration and aging in humans[48] and mice[11]. Additionally, systemic elimination of cells positive for p16INK4a, an important marker of cell senescence in cartilaginous tissues[49], ameliorated disc degeneration in old mice[10]. Despite this causative relationship between senescence and disc pathology, long-term senolytic treatment has not been used to in pre-clinical models of age-associated disc degeneration. We therefore treated BL6 mice weekly with D + Q starting at 6 months (healthy adult disc), 14 months (signs of early disc degeneration), and 18 months of age (established disc degeneration)[6] and analyzed all cohorts at 23 months of age, when advanced disc degeneration is evident. Preservation of tissue and cell morphology and lower grades of degeneration in 6–23 M and 14–23 M, but not 18–23 M, cohorts clearly demonstrated that D + Q treatment slows the progression of disc degeneration if treatment begins in the early stages of the disease process but fails to rescue disc health after degeneration is well-established. This is possibly due to a loss of cellular responsiveness and plasticity required to interfere with the degenerative process. This observation has implications for future translational studies, as a favorable outcome appears to be dependent on identifying the optimal therapeutic window.

The ability of D + Q treatment to reduce the local senescence burden in the disc compartments was tested. In accordance with previous studies of other tissues, D + Q treatment successfully targeted and reduced p16INK4a-, p19ARF-, and p21-positive cells in the intervertebral disc. This may have preserved cellularity as well as enabled a greater abundance of phenotypic markers in D + Q-treated animals, substantiating studies of cartilage by Jeon et al.[20]. Despite the decrease in senescence in the D + Q group, we did not observe any changes in proliferation at 23-months. This could be due to two reasons. First, the proliferation may have occurred at an earlier time, and was therefore not evident at 23-months, and second, in mice the disc progenitor cells decline with aging, leading to decreased proliferation and capacity to renew resident cells[50]. Noteworthy, studies suggest that SASP is the principal mechanism responsible for promoting local fibrosis, degeneration, and inflammation by senescent cells[35,51]. Levels of IL-6 and MMP13, known products of the SASP, were lower in the D + Q-treated group. However, IL-1β levels were higher in the D + Q cohort when compared to the Vehicle but not the 1 y Ctr group[11,35]. While this differed from the analysis of blood in these mice, the effect in the disc was not unexpected, as a recent study by Gorth et al.[52] highlighted the importance of baseline IL-1 expression for the maintenance of disc health during aging[52]. Likewise, a recent study on disc aging showed that IL-1 levels in disc did not correlate with the abundance of senescence markers p21 and p16[11]. Together, these findings suggest that unlike other tissues IL-1 is not a characteristic SASP marker in the disc[35]. Additionally, supporting previous studies of p16INK4a conditional deletion in the adult disc and clearance of p16INK4a-positive cells in human NP cell pellet cultures by RG-7112[10,11,16], D + Q

mitigated levels of inflammatory mediators and metalloproteinases. Importantly, treatment preserved the number of disc cells in part through a reduction in age-dependent cell death and simultaneously maintained NP cell phenotypic marker expression. These results were similar to observations that intra-articular injection of a senolytic drug, UBX0101, preserved the chondrocyte phenotype in a model of surgically induced osteoarthritis[20].

Since alterations in extracellular matrix composition and integrity highly correlate with disc function[7,28], healthy disc matrix (COL 1, COL 2, and CS) and degenerative (COLX and ARGxx) constituents were evaluated in treated mice[37]. While D + Q treatment did not restore all matrix constituents to levels seen in 1 y Ctr animals, there was a noticeable improvement over the Veh group, with higher levels of COL 1 and CS and lower levels of COLX and ARGxx. Patil et al.[10] have shown reduced levels of matrix catabolism in the disc when p16INK4a-positive cells were systemically deleted. Noteworthy, we observed lower incidence of NP fibrosis in the D + Q-treated group. Interestingly, D + Q treatment in a bleomycin-induced idiopathic pulmonary fibrosis mouse model was shown to successfully reduce lung fibrosis by decreasing local senescence burden, resulting in improved health and function[23]. Importantly, our findings of D + Q efficacy in targeting SASP and restoring matrix homeostasis align with recent human trials that showed effectiveness of this drug cocktail in targeting senescence and improving the physical condition of patients suffering from idiopathic pulmonary fibrosis and terminal renal disease[26,27].

Although several studies have reported the potential of D + Q treatment to prevent disease progression through the reduction of senescence and degeneration, the molecular mechanisms modulated by these drugs at the transcriptomic level are unexplored. Aging significantly alters transcriptomic profiles of the intervertebral disc, with a consequent impact on morphological phenotype[28]. While several studies have shown transcriptomic differences between healthy AF and NP cells, during aging and degeneration, the demarcation between these two compartments is lost, and NP cells start to express markers reminiscent of hypertrophic chondrocytes[6,7,53]. Our results clearly showed that each compartment maintained expression of its unique markers in old mice, but differences in cell cycle regulation were evident. This result is of particular interest since D + Q is shown to target a pro-survival network of tyrosine kinases, BCL-2, p53, p21, serpine, PI3K/AKT, and consequently the cell cycle[17,18]. Thus, it was not surprising that a high percentage of genes affected by D + Q treatment were unique to the AF and NP compartments. Importantly, the AF in D + Q mice showed similar increases in DEGs related to response to stress, mitotic cell cycle, response to DNA damage stimulus, and DNA repair as reported in other tissues[17]. Likewise, previous studies with ERCC-/Δ mice showed higher grades of disc degeneration, senescent cells, and matrix catabolism[9,54]. Additionally, downregulated DEGs showed enrichment for several pathways related to cell respiration, suggesting that inhibition of oxidative metabolism is crucial to maintain disc homeostasis[55–57]. In the NP compartment, in agreement with lower histological grades of degeneration and COL X abundance, a marker of hypertrophic chondrocytes[7], downregulated DEGs showed enrichment in skeletal cell differentiation. Noteworthy, NP and AF tissues showed modulation of cell death, response to cytokines along with regulation of RNA polymerase, and the ERK1/ERK2 cascade. These transcriptomic studies are particularly interesting as they underscore common mechanisms regulated by D + Q, independent of the tissue type, and further support TUNEL staining and SASP findings[17,20,24]. Moreover, by comparing DEGs modulated by D + Q treatment to aging-dependent transcriptomic signature, we discovered that

D + Q promoted negative regulation of protein kinase activity (Dasatinib targets senescence by inhibiting tyrosine kinase protein[17]) and promoted cell viability by negatively regulating apoptotic process and modulated NP cell differentiation pathways. Together, these results provide clear evidence that, in addition to tissue-specific effects, D + Q treatment modulates common biological pathways, partially explaining the overall protective effect observed in different organ systems.

Since most studies have studied shorter term D + Q administration, we sought to evaluate the systemic impact of long-term treatment on other organ systems. Previously, D + Q treatment improved vasomotor function by increasing nitric oxide bioavailability and reducing aortic calcification during aging[58]. Analysis of circulatory inflammatory mediators clearly showed a significant decrease in critical proinflammatory proteins and Th17 mediators in all treatment cohorts. Noteworthy, the 6–23 M cohort presented more pronounced downregulation of these factors, suggesting that longer durations of D + Q treatment contributed to better control of systemic inflammation. SASP can promote systemic inflammation and immune system activity[35]. Indeed, some studies have explored the effects of senolytic and senostatic drugs that target SASP secretion to decrease tissue inflammation and enhance the immune system activity[44,51,59]. Noteworthy, the intervertebral disc is an avascular and constrained tissue that exhibits insensitivity to systemic inflammation[60,61]. This fact may explain the differences seen between systemic- and disc-level modulation of IL-6 and IL-1. Despite these differences, it was interesting to observe that D + Q treatment promoted downregulation of proinflammatory signaling in the disc and in circulation. While it is important to acknowledge that the possible systemic effects of D + Q treatment may contribute to ameliorating disc degeneration during aging, senolysis plays a key role in this process since 18–23 M cohort, which did not evidence decreased senescence burden also lacked rescue of aged-associated degeneration. Moreover, along with the beneficial systemic effects of D + Q treatment, we observed an increase in physical strength of mice in the 6–23 M cohort. This was important, since strength is one of the most accurate indicators of systemic aging[62,63], and similar results of improved physical performance and decreased SASP expression in mice treated with D + Q from 20 to 24 months have been noted[25]. Furthermore, recent human trials of idiopathic pulmonary fibrosis using D + Q treatment have shown improvements in patients' physical condition[26,27]. Importantly, long term treatment of mice in our study did not show noxious effects, underscoring the safety of D + Q treatment. Additionally, and in agreement with a previous report, our 18–23 M cohort showed improved vertebral bone quality parameters[24]. Surprisingly, we did not observe the same beneficial effects on vertebral bone quality in 6–23 M and 14–23 M cohorts. Interestingly, a recent study suggested that targeting senescent osteoclasts did not promote increased bone quality in old mice[64]. It is plausible an optimal treatment window is necessary to target-specific senescent populations within bone, affecting bone homeostasis during aging. A tissue that is similar to the intervertebral disc, due to its anatomy and matrix composition, is the articular cartilage. Jeon et al.[20] have previously shown promising results of intra-articular injections of senolytic compound UBX0101 ameliorating osteoarthritis progression in an ACL injury mouse model. In our study, while OA development was seen with aging (especially in male mice), it was surprising that no significant improvements in age-related osteoarthritis were noted in our D + Q treatment cohorts. Since there were no changes in senescence status in cartilage, it is plausible that similar to AF and NP cells, chondrocytes exhibit differential sensitivity and response to D + Q treatment, making drug choice and administration method

important factors while devising senolytic treatments for specific tissues. Indeed, more than 60% of the senescent chondrocytes from within murine cartilage explants were cleared with 3 days of navitoclax treatment[65], but this senolytic was not directly compared to D + Q in that study. Another possibility is that intra-articular injection of D + Q would be needed to obtain sufficient drug concentration in the joint. Importantly, some studies have suggested that different senolytic compounds might present diverse efficacy depending on the target tissue[23,66]. Additionally, based on our results the beneficial effects of systemic administration of D + Q were clearly dependent on the analyzed tissue type and duration of the treatment.

In summary, systemic administration of D + Q[22,67] posits an exciting therapeutic approach to treating disc degeneration, without the inherent risks associated with invasive surgical interventions. The potential benefits of D + Q treatment include alleviation of disc degeneration, reduction in systemic inflammation, and improved physical condition during aging. We also provide insights into tissue-specific effects of D + Q treatment and underscore the importance of treatment duration. These issues will be critical to consider during future pre-clinical studies in large animal models of disc degeneration.

## Methods

**Mice, treatment, and study design**. All animal experiments were performed under Institutional Animal Care and Use Committee (IACUC) protocols approved by the Thomas Jefferson University, and all the guidelines for animal experiments were adhered to. 6-month-old (healthy adult, 15 females, 13 males), 14-month-old (middle-aged, 7 females, 8 males), and 18-month-old (aged, 7 females, 13 males) C57BL/6 mice were obtained from the aged rodent colony at the National Institutes of Aging (NIA), aged to 23 months (old age), and analyzed. These time-points were chosen based on previous studies showing progressive changes in morphological features of mouse lumbar discs at these time-points[6]. The endpoint of 23 months was selected according to guidelines on life phases and maturational rates for the C57BL/6J strain compared to humans[68]. 1-year-old C57BL/6 mice were used as a baseline reference for middle-aged, healthy adult discs to address age-related directionality of changes[69]. Animals were randomly divided into 2 groups per timepoint and received weekly intraperitoneal injections of either Vehicle (1:2 PBS and 1:2 DMSO) or 5 mg/kg Dasatinib (Sigma-Aldrich, CDS023389) plus 50 mg/kg Quercetin (Sigma-Aldrich, Q4951) until they were 23 months old. Senolytic drug concentrations were based on previous studies demonstrating a stronger senolytic effect of D + Q in combination at these concentrations in alleviating the senescence burden in different tissues[17,23]. To ascertain how baseline degeneration status of discs affects the success of the D + Q therapy in ameliorating an age-dependent progression of degeneration, mice were treated before presenting signs of degeneration (6 months), as they evidence early degenerative changes (14 months), and when established age-related degeneration is observed in the disc (18 months)[6]. Mouse weights were recorded weekly, and the number of animals in each cohort was as follows: 6–23 M Veh (n = 13), D + Q (n = 15); 14–23 M Veh (n = 8), D + Q (n = 7); 18–23 M Veh (n = 11), D + Q (n = 9).

**Grip test analysis**. Grip strength was measured using DFIS-2 Series Digital Force Gauge (Columbus Instruments, OH), as reported previously[70]. Briefly, a digital force gauge is attached to a triangular metal pull bar that allows for the transduction of force from the mouse to the gauge. This apparatus measures the strength with which the mouse can grasp and hold on to a thin metal bar. Mice from the 6–23 M cohort were acclimated for a week and tested just before euthanasia. In each trial, the mouse was allowed to grab the bar with one forepaw and was then quickly pulled away from the gauge, so its grip was released, providing a measurement of the force with which the mouse gripped the bar. A blinded examiner performed three trials for each forepaw, with an inter-trial interval of at least 60 s.

**Histological analysis**. Dissected spines were fixed in 4% PFA in PBS for 48 hours, decalcified in 20% EDTA, and embedded in paraffin. Seven micrometer mid-coronal sections were cut from three lumbar levels (L3-6) of each mouse, stained with Safranin-O/Fast Green/Hematoxylin for histological assessment and with Picrosirius Red for collagen fiber characterization. Safranin-O staining was visualized using an Axio Imager 2 microscope (Carl Zeiss) using 5×/0.15 N-Achroplan or 20×/0.5 EC Plan-Neofluar objectives (Carl Zeiss) and Zen2TM software (Carl Zeiss AG, Germany). Mid-coronal sections from ≥3 lumbar discs (L3-S1) per mouse were scored using a modified Thompson grading scale by four blinded observers[7]. The heterogeneity of collagen organization was evaluated using a polarizing, light microscope, Eclipse LV100 POL (Nikon, Tokyo, Japan) with a 10x/0.25 Pol/WD 7.0 objective and DS-Fi2 camera and images analyzed in the NIS

Elements AR 4.50.00 software (Nikon, Tokyo, Japan). Under polarized light, stained collagen bundles appear either green, yellow, or red and correlate to the fiber thickness. Color threshold levels were maintained constant between all analyzed images.

*Imaging FTIR spectroscopy and spectral clustering*. Five micrometer deparaffinized sections of decalcified lumbar disc tissues were collected from vehicle and D + Q 14–23 M cohort group (n = 3 disc/animal, 6 animals/group) and used to acquire FTIR spectral imaging data using methods previously described[40]. Data was collected using Spectrum Spotlight 400 FTIR Imaging system (Perkin Elmer, Shelton, CT), operating in the mid-IR region of 4,000 - 850 cm⁻¹ at a spectral resolution of 8 cm⁻¹ and spatial resolution of 25 μm. Spectra were collected across the mid-IR region of three consecutive sections/disc to minimize section-based variation. Using the ISys Chemical Imaging Analysis software (v. 5.0.0.14) mean second-derivative absorbances in the collagen side-chain vibration (1338 cm⁻¹) regions were quantified and compared in Veh and D + Q AF and NP compartments. The preprocessed spectra were used for K-means cluster analysis to define anatomical regions and tissue types within the tissue section spectral images[7], which represent collagen peak. Clustering images were obtain using Spectrum Image Software.

*Immunohistology and cell number measurements*. Deparaffinized sections following antigen retrieval were blocked in 5% normal serum in PBS-T, and incubated with antibodies against collagen I (1:100, Abcam ab34710), collagen II (1:400, Fitzgerald 70R-CR008), COMP (1:200, Abcam ab231977), collagen X (1:500, Abcam ab58632), chondroitin sulfate (1:300, Abcam ab11570); CA3 (1:150, Santa Cruz), p16 (1:50, Abcam ab211542), p19 (1:100, Novus NB200-106), p21 (1:200, Novus NB100-1941), p-H2AX (1:50, Cell Signaling 9718), RB (1:50, Abcam ab181616), pRB (1:50, Cell Signaling D20B12), Ki67 (1:100, Abcam ab15580), IL-1β (1:100, Novus NB600-633), IL-6 (1:50, Novus NB600-1131), and MMP13 (1:150, Abcam ab39012). For GLUT-1 (1:200, Abcam, ab40084) and ARGxx (1:200, Abcam, ab3773) staining, a MOM kit (Vector laboratories, BMK-2202) was used for blocking and primary antibody incubation. Tissue sections were washed and incubated with species-appropriate Alexa Fluor-594 conjugated secondary antibodies (Jackson ImmunoResearch Lab, Inc.,1:700). F-CHP (3Helix) staining was performed according to the manufacturer's protocol. The sections were mounted with ProLong® Gold Antifade Mountant with DAPI (Fisher Scientific, P36934), visualized with Axio Imager 2 microscope using 5×/0.15 N-Achroplan or 20×/0.5 EC Plan-Neofluar objectives, and images were captured with Axiocam MRm monochromatic camera (Carl Zeiss) and Zen2TM software (Carl Zeiss AG, Germany). Both caudal (Ca5-8) and lumbar (L3-S1) discs were used for the analysis. Staining area and cell number quantification were performed using the ImageJ software, v1.53e, last access 12/05/2021 (http://rsb.info.nih.gov/ij/). The boundaries of the NP and AF were digitally traced using the Freehand Tool. Images with selected ROI (NP, AF, and EP) were thresholded to subtract the background, transformed into binary format, and then staining area and cell number were calculated using the *analyze particle* function in Image J software, v1.53e, last access 12/05/2021 (http://rsb.info.nih.gov/ij/)[56].

*TUNEL assay*. TUNEL staining was performed using the In situ *cell death detection kit* (Roche Diagnostic). Briefly, sections were deparaffinized and permeabilized using Proteinase K (20 μg/mL) for 15 min, and the TUNEL assay was carried out per the manufacturer's protocol. Sections were washed and mounted with ProLong® Gold Antifade Mountant with DAPI and visualized and imaged with the Axio Imager 2 microscope.

*Tissue RNA isolation and real-time RT-PCR analysis*. NP and AF tissues were dissected from lumbar (L1-3) and caudal discs (Ca1-5) of 23-month-old Veh and D + Q animals from the 14–23 M cohort (Veh = 5 and D + Q = 4). Pooled tissue from a single animal served as an individual sample. Samples were homogenized, and total RNA was extracted using the RNeasy® Mini kit (Qiagen). The purified, DNA-free RNA was converted to cDNA using the EcoDry™ Premix (Clontech). Template cDNA and gene-specific primers (IDT, IN) were added to Power SYBR Green master mix, and expression was quantified using the Step One Plus Real-time PCR System (Applied Biosystems).

*Microarray analysis and enriched pathways*. Total RNA with RIN > 4 was used for the analysis. Fragmented biotin-labeled cDNA was synthesized using the GeneChip WT Plus kit according to the ABI protocol (Thermo Fisher). Gene chips (Mouse Clariom S) were hybridized with biotin-labeled cDNA. Arrays were washed and stained with GeneChip hybridization wash & stain kit and scanned on an Affymetrix Gene Chip Scanner 3000 7G, using the Command Console Software. Quality Control of the experiment was performed in the Expression Console Software v 1.4.1.CHP files were generated by sst-rma normalization from Affymetrix. CEL files, using the Expression Console Software. Only protein-coding genes were included in the analyses. Detection above background higher than 50% was used for Significance Analysis of Microarrays (SAM), and the p-value was set at 5%. Biological process enrichment analysis was performed using the PANTHER Overrepresentation Test with GO Ontology database annotations and a binomial statistical test with FDR ≤ 0.05. Analyses and visualizations were conducted in the Affymetrix Transcriptome Analysis Console (TAC) 4.0 software. Pathway

schematic analyses were obtained using the TAC. The array data generated during this study is available in a public repository, GEO database, GSE154619.

To study how D + Q treatment modulated aging related transcriptome changes in NP tissue, we analyzed the array results previously deposited in GEO database: GSE134955, $p < 0.05$ (23 M C57BL/6 mice $n = 7$ vs. 6M C57BL/6 mice $n = 7$)[28] and compared them to D + Q modulated DEGs form this study, GSE154619.

*Micro-CT analysis.* Micro-CT (μCT) scanning (Bruker SkyScan 1275, version 1.0.16.0) was performed on fixed spines using parameters of 50 kV (voltage) and 200 μA (current) at 15 μm resolution. Images were reconstructed using the nRecon program (Version: 1.7.1.0, Bruker) and analysis was performed using CTan (version 1.17.7.2 + 64 bit Bruker micro-CT). Transverse cross-sectional images were analyzed to evaluate trabecular and cortical bone morphology. For trabecular analysis, a region of interest (ROI) was selected by contouring the boundary between trabecular and cortical bone throughout the vertebral body. The 3D datasets were assessed for bone volume fraction (BV/ TV), trabecular thickness (Tb. Th), trabecular number (Tb. N), and trabecular separation (Tb. Sp). For cortical bone analyses, 2D assessments were computed for cortical bone volume (BV), cross-sectional thickness (Cs.Th), polar moment of inertia (MMI), and Eccentricity (Ecc). Disc height and vertebral length were measured at three different points equidistant from the center of the bone on the sagittal plane. Disc height index (DHI) was calculated using equation: Disc Height Index (DHI) $= 2 \times (DH1 + DH2 + DH3)/(A1 + A2 + A3 + B1 + B2 + B3)$ where A and B represent the length of the vertebral bone immediately rostral and caudal to the intervertebral disc, respectively; and DH represents the disc height between adjacent vertebrae[7,71].

*Circulating cytokine analysis.* Blood was collected by heart puncture following sacrifice and centrifuged at $1500 \times g$, at 4 °C for 15 min to isolate the plasma, which was stored at −80 °C until analysis. Levels of proinflammatory proteins, cytokines, and Th17 mediators were analyzed using V-PLEX Mouse Cytokine 29-Plex Kit (Meso Scale Diagnostics, Rockville, MD) and Discovery Workbench 4.0 software according to manufacturer's specifications.

*Statistics and reproducibility.* All statistical analyses were performed using Prism7 (GraphPad, La Jolla). Data are represented as mean ± SEM. Data distribution was assessed with the Shapiro–Wilk normality test, and the differences between the two groups were analyzed by *t*-test or Mann–Whitney, as appropriate. The differences between three groups were analyzed by ANOVA or Kruskal–Wallis for non-normally distributed data, followed by a Dunn's multiple comparison test. A $\chi^2$-test was used to analyze the differences between the distribution of percentages. $p \leq 0.05$ was considered a statistically significant difference. Reported results were consistently replicated across multiple experiments with all replicates generating similar results: 6–23 M cohorts: 4 independent experiments, with 3–4 animals per group (Veh and D + Q); 14–23 M cohorts: 2 independent experiments, with 3–4 animals per group (Veh and D + Q); 18–23 M cohorts: 4 independent experiments, with 2–3 animals per group (Veh and D + Q).

**Reporting summary**. Further information on research design is available in the Nature Research Reporting Summary linked to this article.

## Data availability

The microarray dataset data generated in this study is deposited in the GEO database under accession code GSE154619. There are no access restrictions to any of the processed and raw data. Source data are provided with this paper.

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

## Acknowledgements

This study is supported by the grants from the National Institute of Arthritis and Musculoskeletal and Skin Diseases (NIAMS) R01AR055655, R01AR064733, and R01AR074813 to M.V.R. E.J.N. received a Ph.D fellowship (PD/BD/128077/2016) from the MD/Ph.D Program at the University of Minho, funded by the Fundação para a Ciência e a Tecnologia (FCT). B.O.D. is supported by start-up funds provided by the Thurston Arthritis Research Center and Biomedical Engineering. Support for KDR is provided by 2-T35-AG038047, UNC-CH MSTAR Program under a National Institute on Aging grant.

## Author contributions

E.J.N., B.O.D., I.M.S., and M.V.R. conceived the project. E.J.N., B.O.D., and M.V.R. designed experiments, conducted experiments, analyzed data, wrote the manuscript, and prepared figures. V.A.T., K.R.D., A.J.R., and G.A.S. conducted experiments and analyzed data.

## Competing interests

The authors declare no competing interests.
