## [Peer Review File · Nature Communications]

Long-term treatment with senolytic drugs Dasatinib and Quercetin ameliorates age-dependent intervertebral disc degeneration in micePeer Review File

Reviewer comments, first round –

Reviewer #1 (Remarks to the Author):

NCOMMs-20-48641

“Long-term treatment with senolytic drugs Dasatinib and Quercetin ameliorates age-dependent intervertebral disc degeneration in mice” by Novais et al.

In recent years, it has become clear that the accumulation of senescent cells during the aging process is associated with a variety of harmful side effects, and the development of Senolytic drugs that specifically eliminate senescent cells is anticipated. The combination of dasatinib (D) and quercetin (Q) was the first reported senolytic agent and is currently being tested in clinical trials. Here, the authors explore the ability of different dosing periods of (D+Q) to prevent the progression of age-dependent disc degeneration in C57 BL/6 mice. The authors demonstrated that D+Q treatment slows the progression of disc degeneration if D+Q treatment begins in the early stages of the disease process but fails to rescue disc health after degeneration is well established.

This is a potentially interesting and important work, as the identification of the optimal duration of administration of senolytic drugs is crucial for their clinical application. Overall, the experiments are well organized and data are for the most part solid. However, there are several limitations, which the authors should address before publication.

Critiques:

- 1) In Supplementary Fig.1, the authors reported that mice in 18-23M cohort did not show any significant improvements in disc morphology relative to the Veh group. However, it is unclear whether this is simply because senolysis did not occur under these conditions, or whether senolysis did occur but disc morphology did not improve. Therefore, the author needs to answer this point experimentally.
- 2) Along similar lines, D+Q treatment in the 18-23M cohort appears to worsen disc morphology (Supplementary Fig.1B to D). The authors should discuss this point further.
- 3) In Figure 1F, the authors showed immunofluorescence data using an anti-p16 antibody. However, I strongly feel that it is necessary to verify whether this data really reflects the expression of p16 using tissues of p16-KO mice side-by-side. The reason why we say this is that we have noticed that much of the data reported so far on tissue staining with p16 antibodies is simply detecting noise.
- 4) In Figure 1J, it is difficult to understand why the authors were able to see a decrease in the pRB signal after D+Q treatment. As we know, the activity of pRB is mainly regulated by phosphorylation by CDKs, but not at the level of expression. Thus, it is difficult to understand why the protein level of pRB is reduced by senolysis.
- 5) In Figure 2A, the level of IL-1, one of the most important SASP factors, is significantly increased after treatment with the senolytic drug. If the number of senescent cells is reduced by senolysis, SASP should be reduced and therefore IL-1 should also be reduced. The authors should explain why this is the case.
- 6) The paper is only phenomenological in its entirety and the mechanism by which D+Q produced this effect remains unclear. The authors should discuss more about the molecular mechanisms behind this phenomenon.
- 7) Grosse et al. have recently reported that the removal of p16-expressing cells causes negative effects in mice (Cell Metab. 2020 Jul 7;32(1):87-99.e6. doi: 10.1016/j.cmet.2020.05.002.). It is

therefore not clear whether the effect of D+Q in this paper is purely an effect of removing senescent cells or whether it reflects the senescence-independent effect of D+Q. Thus, the authors should use other senolytic drugs to see if they have the same effect as that obtained with D+Q. It should be noted that the efficacy of D+Q as a senolytic drug is lower than that of other senolytic drugs in human fibroblasts (see Wakita et al. Nat Commun 2020 /doi.org/10.1038/s41467-020-15719-6 Supplementary Fig.1). For this reason, other senolytic drugs should definitely be used (Di Micco. Nat Rev MCB 2020 <https://doi.org/10.1038/s41580-020-00314-w>)

Reviewer #2 (Remarks to the Author):

Cellular senescence is a major contributor to aging and age-related disease, including disc degeneration. Earlier studies reported that clearance of senescent cells with oral administration of D+Q improved overall physical function and survival of naturally aged mice. Other groups also reported treating disc degeneration with senolytics. This study by Novais et al tested the efficacy of Dasatinib and Quercetin on preventing age-related disc degeneration. B6 mice received a weekly injection of D+Q beginning at 6, 14, and 18 months of age and analyzed at 23 months. It is not surprising that D+Q lowered the incidence of disc degeneration and suggested that senolytics may halt the age-dependent disc degeneration. The question remains whether this benefit is due to the overall improved health spin of D + Q, as this cocktail has been shown to attenuate the harmful effects of senescence in multiple tissue types. The substantially reduced systemic proinflammatory cytokine following D+Q may also decrease the incidence of disc degeneration. As discs are an avascular organ, local drug concentration may low by the systemic delivery of medications. This is a well-designed and presented study. It is significant because there is no effective drug available to treat age-related disc degeneration.

Some specific comments:

1. Have the authors observe any side effects of D+Q for a long-term (15 months) injection?
2. Earlier studies have shown that senescent may be beneficial to wound healing. Would long-term take of D+Q impact tissue regeneration?
3. Please explain why the histology scores of L4-5 and L5-6 of NP showed lower grade, while AF showed lower scores in L3-5 and L5-6 of AF in the 14-23M cohort but not the same level as NP.
4. Is there a significant difference in disc degeneration between 12 months and 14 months of discs? Why used 1-year-old mice as the controls for the 14-23M cohort? Were there any differences in AF cellularity and % of apoptotic cells in the 14-23M cohort vs vehicle groups?
5. It is interesting that this study did not observe changes in scores of arthritis because joints and intervertebral discs share many features.
6. Were tail discs or lumbar discs used for Histology/immunostaining and gene analysis? It was not clearly stated.

Authors Response

We thank the reviewers for their encouraging comments and valuable feedback to improve the manuscript. We provide point-by-point responses to each reviewer below. Within this response document we used bold font to indicate new data.

Reviewer 1:

Long-term treatment with senolytic drugs Dasatinib and Quercetin ameliorates age-dependent intervertebral disc degeneration in mice" by Novais et al.This is a potentially interesting and important work, as the identification of the optimal duration of administration of senolytic drugs is crucial for their clinical application. Overall, the experiments are well organized, and data are for the most part solid.

The authors thank the reviewer for the encouraging comments and for recognizing the importance of our study.

1. *In Supplementary Fig.1, the authors reported that mice in 18-23M cohort did not show any significant improvements in disc morphology relative to the Veh group. However, it is unclear whether this is simply because senolysis did not occur under these conditions, or whether senolysis did occur but disc morphology did not improve. Therefore, the author needs to answer this point experimentally.*

The reviewer makes an important point. We have now added additional analysis of p16, p19, and RB levels in the 18-23M cohort (**Suppl. Figure 2E-G**). These results showed that the administration of D+Q treatment starting at 18M did not decrease senescence burden in the intervertebral disc during aging. These results strongly support the hypothesis that decreasing disc senescence status is essential to relieve the degenerative phenotype during aging.

2. *Along similar lines, D+Q treatment in the 18-23M cohort appears to worsen disc morphology (Supplementary Fig.1B to D). The authors should discuss this point further.*

We thank the reviewer for this comment. In order to evaluate the possible trend suggested by the reviewer – worsening of disc degeneration in the 18-23M D+Q group - we performed an effect size analysis followed by a power analysis to estimate the sample size needed to reach significance. Thus, considering the 18-23M cohort (Veh = 43 discs, D+Q=36 discs), the mean group of Veh – 2.713 and D+Q – 2.919, and population standard deviation of 0.848, the calculated effect size was 0.243, which is considered small. Likewise, assuming this effect size (0.243), $\alpha = 0.05$, and power of 95%, the sample size needed to detect a statistical difference would be 215 discs per group. In summary, this analysis clearly shows that the effect of D+Q on disc degeneration phenotype in the 18-23M cohort during aging, if present, is minimal. Additionally, we have now stained discs of mice from 18-23 M cohort for senescence markers p16, p19, and RB and found that D+Q treatment, in contrast to the 14-23M cohort, did not ameliorate senescence status. This data further supports the importance of study and points out the best window of treatment to guarantee the highest efficacy and beneficial effect.

3. *In Figure 1F, the authors showed immunofluorescence data using an anti-p16 antibody. However, I strongly feel that it is necessary to verify whether this data really reflects the expression of p16 using tissues of p16-KO mice side-by-side. The reason why we say this is that we have noticed that much of the data reported so far on tissue staining with p16 antibodies is simply detecting noise.*

The reviewer makes an important suggestion. We have tested the specificity of the p16 staining in the intervertebral disc using a mouse model of p16Ink4a conditional deletion in the disc driven by Acan-CreERT2 (p16cKO), previously reported by our group (Novais *et al.* Matrix Biol. 2019). As reported earlier, the Acan-CreERT targets the NP, AF and cartilaginous endplate. Noteworthy, the p16Ink4a cKO mice, similar to the negative control, did not show any quantifiable p16 staining (**Figure shown below**). It is also important to point out that, using this antibody we did not observe any differences in the levels of p16 abundance between Veh and D+Q treated mice from 18-23M cohort, contrasting the findings of 14-23M group (**Suppl. Figure 2E-E''**).

4. *In Figure 1J, it is difficult to understand why the authors were able to see a decrease in the pRB signal after D+Q treatment. As we know, the activity of pRB is mainly regulated by phosphorylation by CDKs, but not at the level of expression. Thus, it is difficult to understand why the protein level of pRB is reduced by senolysis.*

The authors thank the reviewer for pointing this out. To further explore the changes seen in pRB levels, we have now studied the expression levels of RB, pH2AX, and Ki67. Interestingly, despite the decrease seen in p16, p19, pRB, and SASP levels in the D+Q treated mice compared to Veh treated mice, we did not observe any quantifiable expression of Ki67 in both Veh and D+Q groups at 14-23M. This data suggests that amelioration of senescence status with aging did not promote disc cell proliferation at 23M (**Figure 1J-J'**). Similarly, RB and pH2AX levels were maintained in both groups (**Suppl. Figure 2A-D**). Thus, our results suggest that D+Q treatment reduces senescence markers and alters regulation of cell cycle-associated proteins without promoting cell proliferation in 23M old mice. Importantly, our microarray data further supports these results, showing enrichment for cell cycle-related pathways in both D+Q

AF and NP tissues: upregulated DEGs in D+Q AF - mitotic cell cycle; downregulated DEGs in NP – skeletal muscle cell differentiation. We have carefully discussed these findings in the revised manuscript (**Line 394-398**).

5. *In Figure 2A, the level of IL-1, one of the most important SASP factors, is significantly increased after treatment with the senolytic drug. If the number of senescent cells is reduced by senolysis, SASP should be reduced and therefore IL-1 should also be reduced. The authors should explain why this is the case.*

The reviewer highlights an interesting point. Indeed IL-1 is one well-described marker of senescence-associated secretory phenotype (SASP) in different cell types/tissues. However, in disc cells, IL-1 does not seem to be part of canonical SASP signature. We have previously described the senescence makers in C57BL6 mice during aging and, despite an overall increase of p16, p21, and IL-6 levels in old mice, we did not observe any change in the levels of IL-1 (Novais *et al.* Matrix Biology 2019). These results clearly show that IL-1 levels were not associated with an increase in other senescence markers.

Moreover, Gorth *et al.* (J. Bone Miner Res. 2019) have characterized the disc phenotype of IL-1 knockout mice, which showed slightly higher grades of degeneration than the wild-type mice with aging, suggesting that IL-1 signaling is essential for maintaining disc tissue homeostasis during aging. Noteworthy, IL-6 and metalloproteases such as MMP13 are also described as critical players of SASP. In our study, we clearly show that abundance of both these molecules increased during aging in disc cells in parallel with senescence markers. We have elaborated on these findings in discussion (**Line 405-408**).

6. *The paper is only phenomenological in its entirety and the mechanism by which D+Q produced this effect remains unclear. The authors should discuss more about the molecular mechanisms behind this phenomenon.*

The authors agree with the reviewer. We have now added new data characterizing the matrix homeostasis and turn-over in both Veh and D+Q treated groups and characterized the senescence status of the 18-23M group. Importantly, our new data clearly shows a decrease in NP tissue fibrotic processes in the D+Q group using imaging-FTIR and Picrosirius red/polarized microscopic analyses (**Figure 4A-D**). Previous studies have suggested that tissue fibrosis is partially responsible for compromised disc function and is strongly associated with decreased cell viability (Choi *et al.* Matrix Biol, 2018, Zhang *et al.* Matrix Biol. 2018). Importantly, our new results show that the 18-23M D+Q group, which did not improve degeneration with aging, evidence comparable levels of senescence markers to vehicle group. These results suggest that senescence status played an important role in the improvements seen in the 14-23M groups. Finally, by comparing the NP transcriptomic signature of aging NP (23M vs 6M) to that of D+Q treated NP tissue (D+Q vs Veh, 14-23M cohort), we discovered that D+Q promoted negative regulation of protein kinase activity (Dasatinib targets senescence by inhibiting tyrosin kinase), promoted cell viability by negatively regulating apoptotic process and modulated NP cellular differentiation pathways. Interestingly, we also found 27 upregulated and 19 downregulated DEGs with aging that presented opposite modulation after D+Q treatment. These new findings add insights to the molecular pathways and mechanisms that underline beneficial effects of D+Q on disc health. Accordingly, we have elaborated on the molecular mechanisms behind this D+Q mediated improvement in disc health in the revised discussion section (**Line 422-425, Line 451-458 and Line 476-479**).

7. *Grosse et al. have recently reported that the removal of p16-expressing cells causes negative effects in mice (Cell Metab. 2020 Jul 7;32(1):87-99.e6. doi: 10.1016/j.cmet.2020.05.002.). It is therefore not clear whether the effect of D+Q in this paper is purely an effect of removing senescent cells or whether it reflects the senescence-independent effect of D+Q. Thus, the authors should use other senolytic drugs to see if they have the same effect as that obtained with D+Q. It should be noted that the efficacy of D+Q as a senolytic drug is lower than that of other senolytic drugs in human fibroblasts (see Wakita et al. Nat Commun 2020 /doi.org/10.1038/s41467-020-15719-6 Supplementary Fig.1). For this reason, other senolytic drugs should definitely be used (Di Micco. Nat Rev MCB 2020 https://doi.org/10.1038/s41580-020-00314-w).*

We thank the reviewer for this comment and agree that it is hard to differentiate solely between the effect of D+Q in removing senescent cells from other potential beneficial effects. Importantly, our new results describing the senescence status in the 18-23M group that did not present any improvements suggest that removing senescent cells is important to rescue the degenerative phenotype with aging. Likewise, work from Patil *et al.* (Aging Cell 2020), clearing p16 positive cells presents similar rescue as our D+Q treatment. Moreover, analysis of aging associated DEGs modulated by D+Q targeted negatively protein kinases activities, which was described by Zhu *et al.* (Aging Cell 2015) as one of the primary pathways underlying Dasatinib's senolytic action, and promoted cell viability. Despite this evidence, it is also important to acknowledge that D+Q treatment resulted in systemic changes, such as the decrease in cytokines and proinflammatory protein levels in the serum, which can also contribute to a certain extent in alleviating intervertebral disc degeneration. We have therefore acknowledged the possible systemic effect of D+Q treatment and revised our discussion that now includes our new data describing the senescence status in the 18-23M group (**Line 476-479**). We have also included a graphical summary with the main molecular mechanisms underlining the phenotypic rescue (**Figure 9**).

We also observed that the beneficial effect of D+Q treatment is dependent on the tissue analyzed (serum, bone, knee cartilage, annulus fibrosus, and nucleus pulposus) and window of treatment. Likewise, it is not surprising that different senolytic drugs will present different efficacy in targeting senescence depending on the tissue. Indeed Shaffer *et al.* (Nat Commun. 2017), has shown that in model of lung fibrosis, D+Q treatment showed comparable efficiency in removing senescence to a genetic mouse model. Moreover efficacy of D+Q in this lung fibrosis setting was higher than treatment with Navitoclax, another well-described senolytic drug. Moreover, D+Q treatment is now adopted for clinical trials and has shown clear improvements in physical condition in late-stage lung fibrosis patients (Justice *et al.* EBioMedicine 2019) and efficacy in clearing senescence cells in late-stage kidney disease (Hickson *et al.* EBioMedicine 2019).

We agree that it would be interesting to test another senolytic drug compound along with D+Q, although this was not the project's initial goal which was to test the potential of long-term treatment with D+Q in preventing disc degeneration. We have discussed the potential of other senolytic drug options and their possible role in preventing degeneration in the different tissues explored (**Line 504-507**). Importantly, due to the need for long-term treatment to observe the beneficial effects of the senolytic treatment in rescuing age-associated disc degeneration (6-23M and 14-23M - i.e. 9-17 months of continuous treatment), and the extent of resources involved, and the continued COVID impact on research, we are not able to test the effect of a different senolytic drug compound on intervertebral disc degeneration during aging in this study. We acknowledge and expect that due to high level of interest in field of senolytics, new classes of drugs/molecules will continue to be investigated and some could be superior for disc health as compared to D+Q. Noteworthy, during this study, we have tested three different treatment regimens, starting points

(6-, 14- and 18-months) and included analysis of different tissues (NP, AF, vertebral bone, serum, articular cartilage) as well as different frailty indicators (survival curve, grip strength, and body weight), which strongly supports and explores the potential of D+Q treatment in alleviating disc degeneration, without presenting noxious side-effects during long-term treatment. We see these results as an important first step for this therapeutic paradigm that extends from previous studies using genetic models for senescence clearance or short-term treatment in accelerated aging/disc degeneration models.

Reviewer 2:

Cellular senescence is a major contributor to aging and age-related disease, including disc degeneration.....This is a well-designed and presented study. It is significant because there is no effective drug available to treat age-related disc degeneration.

The authors thank the reviewer for recognizing the novelty and significant clinical relevance of this study since, despite the high prevalence and costs associated with disc degeneration, the treatments available are still very limit and not well studied.

1. Have the authors observe any side effects of D+Q for a long-term (15 months) injection?

We thank the reviewer for asking this question. In order to evaluate potential deleterious effects associated with prolonged administration of D+Q treatment, we evaluated several well-described frailty indicators (survival curve, weight, and grip strength) and health of musculoskeletal tissues (articular cartilage and bone) besides intervertebral disc. Interestingly, and despite the differential beneficial effects of D+Q treatment on each tissue, we did not observe any deleterious effects of this long-term treatment. Noteworthy, grip strength, a widely accepted indicator of overall health condition and frailty, was increased in the D+Q treated group in 6-23M cohort (**Suppl. Figure 6G**). Additionally, all three treatment cohorts showed a decrease in serum inflammation markers (**Figure 7 and Suppl. Figure 5**). Lastly, the survival curves were comparable between treated and vehicle groups (**Figure 8A-D and Suppl. Figure 6**). These results suggest an overall beneficial effect of D+Q injections without noticeable deleterious effects.

2. Earlier studies have shown that senescent may be beneficial to wound healing. Would long-term take of D+Q impact tissue regeneration?

We thank the reviewer for the comment. Since the scope of this study was to study the effect of long-term treatment of D+Q for ameliorating age-associated intervertebral disc degeneration, it is difficult to speculate about the effect of this treatment on wound healing. Although we did not test the ability of treated mice to regenerate skin or cartilage in response to an intentional wounding model, we did not observe obvious defects in the skin or changes in the hair at any time point during the natural course of aging and response to any wounds that occurred during normal cage behavior (**See Figure below**).

3. *Please explain why the histology scores of L4-5 and L5-6 of NP showed lower grade, while AF showed lower scores in L3-5 and L5-6 of AF in the 14-23M cohort but not the same level as NP.*

We thank the reviewer for pointing this trend. As suggested by different studies, NP and AF tissues can directly and indirectly influence each other's degeneration process by modulating local mechanics and molecular homeostasis. This becomes more evident at higher grades of degeneration (Grade 4-5), but both tissues can also present independent deterioration, more relevant for medium grades of degeneration (Grade 3), similar to the average grades seen in the Veh group. Noteworthy, we have included these references in the introduction section to further elaborate on the interaction between AF and NP degenerative processes (**Line 64-65**).

4. *Is there a significant difference in disc degeneration between 12 months and 14 months of discs? Why used 1-year-old mice as the controls for the 14-23M cohort? Were there any differences in AF cellularity and % of apoptotic cells in the 14-23M cohort vs. vehicle groups?*

As reported recently by Ohnishi et al. (J. Orthop Res 2018), 14-month mice can show some signs early disc degeneration. Our extensive experience with scoring disc degeneration of C57BL/6 mice at our institution gave us confidence that 12-month mice would consistently show healthy discs. Mice at 1 year are mature and equivalent to “middle age” and therefore make a good control to study and compare age-associated differences.

As shown in the **Figure 2F-F** there was an overall increase in TUNEL positive cells in the disc (NP and AF combined). Figures shown below (A and B), clearly show that there is an increase in cell death in both the NP and AF compartments with aging, as seen from higher staining and reduction in cells (DAPI positive nuclei). Importantly, D+Q treatment improved disc cell viability by increasing cellularity in the NP and improved cell viability in the AF. Noteworthy, these results further support the changes seen in the transcriptomic experiment which shown that D+Q modulated DEGs associated with cell viability and negative regulation of apoptosis pathways (**Figure 5J and 6C**).

A

B

5. It is interesting that this study did not observe changes in scores of arthritis because joints and intervertebral discs share many features.

We agree with the reviewer's assessment that this is an interesting finding. However, this was not altogether surprising since different studies have shown that both intervertebral disc and knee joints present unique molecular, mechanical, and vascular features. Previous studies showing a beneficial effect of senescence clearance for the joint have used direct intra-articular injection (Faust *et al.* J Clin Invest,

2020). This may indicate the need for higher concentrations of drug locally to promote the desired effect. Finally, each senolytic compound may present different efficacy depending on the tissue being evaluated, as shown previously by Shaffer *et al.* (Nat Commun, 2017 - D+Q has higher efficacy than Navitoclax in treating lung fibrosis) and Wakita *et al.* (Nat Commun 2020 - BET family protein degradation showed the highest efficiency in IMR90-ER: HRAS cell line). Although we did not observe reduction in senescence status in the knee articular cartilage after D+Q treatment, UBX0101 (Jeon *et al.* Nat Commun. 2017) and Navitoclax (Sessions *et al.* FASEB J. 2019) have shown promising results in targeting the knee joint and articular cartilage explants, respectively. We have further discussed the differential effects of D+Q depending on the tissues and added some new references to comment on the different efficacy of senolytic drugs depending on the tissue type (**Line 504-507**).

6. *Were tail discs or lumbar discs used for Histology/immunostaining and gene analysis? It was not clearly stated.*

We thank the reviewer for this comment and regret the confusion caused. The grading was performed only using lumbar discs since in humans' lumbar discs are the ones that show the highest prevalence of disc degeneration. Due to the limited amount of tissue to perform the remaining experiments, we have then use both the lumbar and tail discs (same levels for each group Veh and D+Q) to perform both immunostainings and gene expression analysis. We have now clarified all the details about the levels used for each experiment in the methodology section.

Reviewer comments, second round –

Reviewer #1 (Remarks to the Author):

I have read the revised manuscript and found that the authors have adequately addressed all of my concerns. Therefore, I support the publication of the revised manuscript in Nature Communications.

Reviewer #2 (Remarks to the Author):

The authors addressed prior concerns.